# Detecting single gravitons with quantum sensing

Germain Tobar [1,2,5], Sreenath K. Manikandan [3,5], Thomas Beitel[4] & Igor Pikovski [1,4] ✉

The quantization of gravity is widely believed to result in gravitons – particles of discrete energy that form gravitational waves. But their detection has so far been considered impossible. Here we show that signatures of single graviton exchange can be observed in laboratory experiments. We show that stimulated and spontaneous single-graviton processes can become relevant for massive quantum acoustic resonators and that stimulated absorption can be resolved through continuous sensing of quantum jumps. We analyze the feasibility of observing the exchange of single energy quanta between matter and gravitational waves. Our results show that single graviton signatures are within reach of experiments. In analogy to the discovery of the photo-electric effect for photons, such signatures can provide the first experimental clue of the quantization of gravity.

Merging Einstein's theory of gravity and quantum mechanics is one of the main outstanding problems of modern physics. A major challenge is the lack of experimental evidence for quantum gravity, with few known experimentally feasible goals. Apart from possible signatures from cosmological observations[1], the advent of quantum control over a variety of quantum systems has enabled searches also in laboratory experiments at low energies[2–11]. This research is fueled by increasing mastery over quantum phenomena at novel mass-scales, such as matter-waves with large molecules[12], quantum control over opto-mechanical systems[13,14], or demonstration of quantum states of macroscopic resonators[15,16]. Proposals with these and similar new quantum systems focus mainly on tests of phenomenological models of quantum gravity that result in modifications to known physics[2–4,17]. Tests of quantum phenomena stemming from gravity within expected physics have been proposed for large superpositions of gravitational source masses[6–8] or quantum noise from gravitons[18,19], but these are far outside the reach of current experiments. The detection of single spin-2 gravitons—the most direct evidence of quantum gravity—has so far been considered a near impossible task[20–23].

Here we show that signatures of single gravitons from gravitational waves can be detected in near-future experiments, in essence, through a gravito-phononic analog of the photo-electric effect and continuous quantum measurement of energy eigenstates. We study the interaction of gravitational waves with quantum matter and show that novel bar resonators cooled to their quantum ground state can, in principle, detect single gravitons. This ability stems from a combination of new experimental developments and theoretical insights: (i) The weak coupling of gravitational waves plays to our advantage: Because the interaction strength between gravitational waves and matter is so small, out of the ~$10^{36}$ gravitons interacting with the detector only very few gravitons end up being absorbed. (ii) Experimental advancements are reaching the ability to prepare quantum states of internal modes of massive systems and to measure them with high precision in time-continuous non-destructive measurements. (iii) One can now correlate events with independent classical detections such as at LIGO for a heralded signal. Together these capabilities open the door to measure single gravitons with gravitational wave detectors deep in the quantum regime, as we show in this article.

[1]Department of Physics, Stockholm University, SE-106 91 Stockholm, Sweden. [2]Okinawa Institute of Science and Technology, 1919-1 Tancha, Onna-son, Kunigami-gun, Okinawa 904-0495, Japan. [3]Nordita, KTH Royal Institute of Technology and Stockholm University, SE-106 91 Stockholm, Sweden. [4]Department of Physics, Stevens Institute of Technology, Hoboken, NJ 07030, USA. [5]These authors contributed equally: Germain Tobar, Sreenath K. Manikandan. ✉e-mail: pikovski@stevens.edu

## Results

### Absorption and emission rates of gravitons

In general relativity, gravity interacts via the non-linear Einstein equations that connect space-time geometry to matter. In the weak field limit of gravity, the equations simplify, where we can approximate the metric as $g_{\mu\nu} \approx \eta_{\mu\nu} + h_{\mu\nu}$, the sum of the flat Minkowski metric $\eta_{\mu\nu}$ and a small perturbation $h_{\mu\nu}$. To first order in $h_{\mu\nu}$, the linearized equations allow for wave-solutions, which are the gravitational waves (GWs) as confirmed by LIGO[24]. The coupling to matter in this linearized limit can be described by the Hamiltonian

$$H_{\text{int}} = -\frac{1}{2} h_{\mu\nu} T^{\mu\nu}, \tag{1}$$

where matter interacts with the gravitational field $h_{\mu\nu}$ through the stress–energy tensor $T_{\mu\nu}$. Eq. (1) is a convenient description for our purposes as it can be readily quantized in both the gravitational and the matter degrees of freedom. The quantized linearized theory of gravity was already considered in 1935 by Bronstein[25], and it results in gravitons in direct analogy to photons in electromagnetism. It is well understood theoretically, as opposed to attempts to quantize the full non-linear theory of gravity. But no experimental evidence of the quantum theory, even in the linearized regime exists to date.

It is easy to see the difficulty if we consider Eq. (1) for a typical quantum system, an atom. Using perturbation theory, we can compute the rate at which gravitons are emitted by an atom using Fermi's Golden rule

$$\Gamma_{\text{spon}} = \frac{2\pi}{\hbar^2} |\langle f | \hat{H}_{\text{int}} | i \rangle|^2 D(\omega), \tag{2}$$

where $\hat{H}_{\text{int}}$ is the interaction Hamiltonian in Eq. (1) with both matter and gravity quantized and $D(\omega) = \frac{V\omega^2}{2\pi^2 c^3}$ is the graviton density of states at frequency $\omega$ and a characteristic volume $V$. This calculation was done by Weinberg[20], and for the quadrupole transition $3d \rightarrow 1s$, one gets $\Gamma_{\text{spon}} \approx 10^{-40}$ Hz[20–22]. Such a low rate is impossible to observe. While some other atomic states, such as Rydberg atoms[26] can provide some enhancement, the rate remains minute. In a similar spirit, Dyson calculated the sensitivity needed to detect a single graviton in LIGO: Currently detected GWs consist of $\gtrsim 10^{36}$ gravitons, as summarized in the "Methods" section, thus measuring a GW consisting of a single graviton would require a position resolution far below the Planck-length. Dyson thus conjectured that it may never be possible to detect gravitons[23].

In contrast, here we now show that signatures of single gravitons can, in fact, become observable, even in near-future experiments. We show that two enhancement mechanisms drastically change the above reasoning: we focus on stimulated processes in a gravitational wave background that can induce single graviton transitions, and we consider massive quantum systems that can be prepared in quantum states at macroscopic mass scales, combined with continuous measurements of single energy quanta.

Starting with Eq. (1), we derive in the "Methods" section the full Hamiltonian that describes gravitons interacting with a collective system of $N$ atoms, which matches previous results in the appropriate limits[27,28]. The dominant interaction with the $l$th odd-numbered mode of a cylindrical resonator with creation (annihilation) operator $\hat{b}_l^\dagger$ ($\hat{b}_l$), frequency $\omega_l$, total resonator mass $M$, effective mode mass $m_{\text{eff}} = M/2$ and length $L$ is

$$\hat{H}_{\text{int},l} = \frac{L}{\pi^2} \sqrt{\frac{M\hbar}{\omega_l}} \frac{(-1)^{\frac{l-1}{2}}}{l^2} \left( \hat{b}_l + \hat{b}_l^\dagger \right) \ddot{\hat{h}}, \tag{3}$$

where $\hat{h}$ is the quantized metric perturbation perpendicular to the resonator (here, for simplicity, we assume a cylindrical resonator and neglect polarization and antenna pattern functions, but a generalization to other geometries and polarizations is straightforward). The semi-classical limit, $\hat{h}$ is just the metric perturbation $h$, and the Hamiltonian becomes the gravitational analog of the quantum optical Rabi Hamiltonian but with the matter system being a harmonic oscillator. This is sufficient to derive our results just as in the photo-electric case, but single transitions of the matter will indicate gravitons by energy conservation. Nevertheless, it is also instructive to consider the full quantum mechanical Hamiltonian[29–31] for which $\hat{h} = \sum_{\mathbf{k}} h_{q,\mathbf{k}} \left( \hat{a}_{\mathbf{k}} + \hat{a}_{\mathbf{k}}^\dagger \right)$, where $h_{q,\mathbf{k}} = \frac{1}{c} \sqrt{\frac{8\pi G\hbar}{V\nu_{\mathbf{k}}}}$ and $\hat{a}_{\mathbf{k}}(t)$ ($\hat{a}_{\mathbf{k}}^\dagger(t)$) are the annihilation (creation) operators for single gravitons with wavenumber $\mathbf{k}$ and frequency $\nu_{\mathbf{k}}$, satisfying the dispersion relation $\nu_{\mathbf{k}} = c|\mathbf{k}|$. This is used in "Methods" section to compute the absorption and emission rates from first principles, as done for the atomic case originally in ref. 22.

We now consider spontaneous emission, as well as stimulated emission and absorption for the macroscopic resonator as described above. Using the interaction Hamiltonian in Eq. (3) and considering the transition of the fundamental resonator mode from the $n = 1$ excited Fock state into the ground state, we obtain the spontaneous emission rate

$$\Gamma_{\text{spon}} = \frac{8GML^2\omega_l^4}{l^4\pi^4 c^5} = \frac{8\pi G\rho\nu_s^4 R^2}{Lc^5}, \tag{4}$$

where $\nu_s = L\omega_l/(l\pi)$ is the speed of sound, $\rho$ the mass density, and $R$ the radius of the cylinder. For a niobium bar of density $\rho = 8570$ kg/m$^3$, speed of sound $\nu_s \approx 0.5 \times 10^4$ m s$^{-1}$, length 1 m, and radius $R = 0.5$ m, we get $\Gamma_{\text{spon}} \approx 10^{-33}$ Hz. This is orders of magnitude better than the results for a single atom as discussed above[20–22], because the rate scales with the macroscopic mass of the system. However, the spontaneous emission rate is still vanishingly small. The spontaneous rate thus remains far beyond possible experimental verification. Nevertheless, this result already highlights two important points: quantum systems that operate on novel scales can bridge many orders of magnitude for testing quantum gravity, and the resulting quantum gravitational effect (the spontaneous emission requires a quantum description of the gravitational field) proceeds on scales far closer than the Planck-scale.

We now show that the stimulated emission and absorption open the door for realistic experiments to detect transitions due to single gravitons. Using again Eq. (3) we obtain the rate of the stimulated transition of the resonator from the ground state $|0\rangle$ to the energy state $|1\rangle$ (and the same rate for the inverse process):

$$\Gamma_{\text{stim}} = \frac{ML^2\omega_l^2 h^2}{4l^4\pi^5\hbar} = \frac{\nu_s^2}{4l^2\pi^3\hbar} M h^2. \tag{5}$$

For a given material the rate thus depends only on the total mass $M$ and is highest for the fundamental mode. The single transition in energy states we consider here corresponds to the absorption or emission of a single graviton by energy conservation. Remarkably, this rate (5) of the exchange of single quanta between matter and gravitational waves can be large: Using an aluminum bar of mass 1800 kg, a gravitational wave with amplitude $h = 5 \times 10^{-22}$ will result in $\Gamma_{\text{stim}} \approx 1$ Hz. Thus, stimulated processes proceed fast enough that single gravitons are exchanged on reasonable time scales, but slow enough that single transitions could be resolved. More generally, in the next section, we derive the exact excitation probability which can be applied to various systems and situations, such as inspirals in the LIGO sensitivity band. Combined

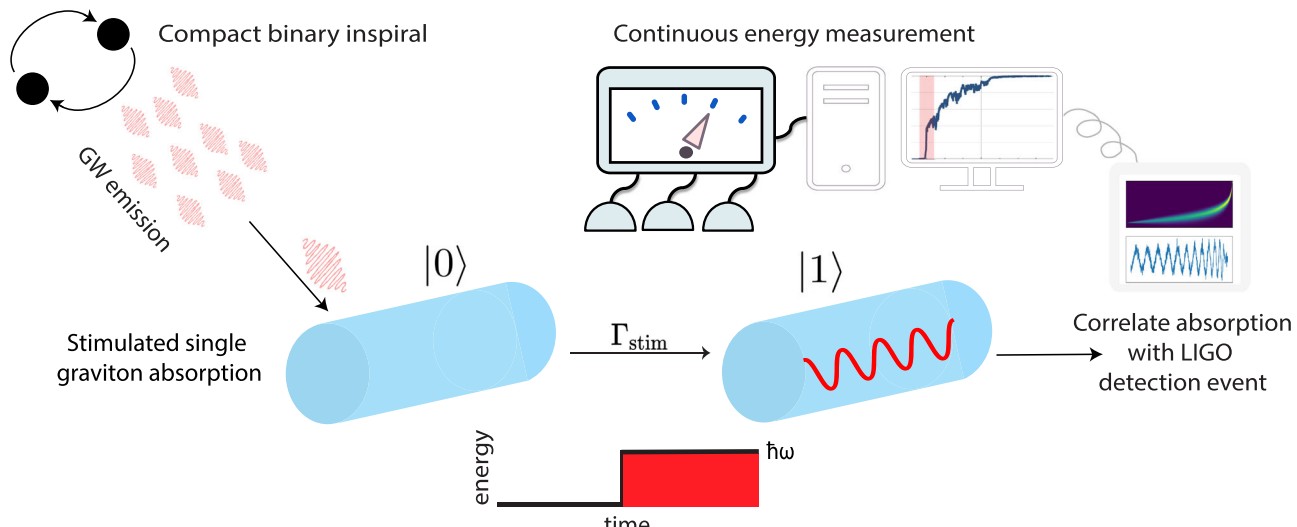

**Fig. 1 | Passing gravitational waves result in stimulated absorption of gravitons.** Due to very low interaction strength, this can be used to design a gravito-phononic analog of the photoelectric effect with acoustic resonators to detect single gravitons. The resonator is cooled to the ground state and its first excited energy level is weakly monitored through continuous quantum sensing. A quantum jump from the ground state to the first excited state corresponds to a single-graviton detection event. Ideal system parameters for such events are given in Table 1. For GWs in the LIGO sensitivity window, correlating the signal to the classical LIGO detection provides confirmation of a single graviton absorption from an incident gravitational wave.

with quantum measurement to continuously monitor quantized energy levels, a signature of a single graviton absorption can be achieved, schematically shown in Fig. 1. We discuss and compute the requirements in more detail below. Assuming only regular quantum mechanics for matter and energy conservation, such an observation would constitute the gravito-phononic analog of the photo-electric effect, historically the first indication of the quantization of light.

## Single graviton signals

Despite the strong classical gravitational wave background, due to its exceptionally weak interaction with matter one can find an experimental regime where only a single graviton is exchanged with a near-resonant mass. We consider three different domains of gravitational waves: compact binary mergers in the LIGO frequency band, continuous waves from neutron stars in the kHz-range, and hypothesized ultra-high frequency gravitational wave signals. Observations in these three domains vary in experimental difficulty, with different parameters that need optimization. A detection in the LIGO band would provide the most likely evidence of gravitons due to the ability to correlate to independent LIGO detections.

   The main goal is to operate a massive bar resonator that could detect the absorption of a single graviton from compact binary mergers. These are detected and confirmed by LIGO[24,32] and thus allow for a correlation measurement between stimulated graviton signals and the detection of classical events. In this case the excitations in the bulk resonator are not well captured by the approximate rate in Eq. (5), which assumes resonance over a very long time. Instead, here we fully solve the dynamics and transition probability induced by the Hamiltonian (3). For simplicity we take the semi-classical limit $\hat{h} \approx h(t)$, neglecting the quantum fluctuations and assuming the GW to be in a coherent state of high amplitude, but from which single gravitons are absorbed. The dynamics can be solved exactly (see the "Methods" section). For its fundamental mode, it is captured by the unitary evolution operator $\hat{U}(t) = e^{-i\varphi(t)} e^{-i\omega t \hat{b}^\dagger \hat{b}} \hat{D}(\beta(t))$ where $\hat{D}(\beta) = e^{\beta \hat{b}^\dagger - \beta^* \hat{b}}$ is the displacement operator with strength $\beta(t) = -i\frac{L}{\pi^2}\sqrt{\frac{M}{\hbar\omega}}\int_0^t ds\,\ddot{h}(s)e^{i\omega s}$ and $\varphi(t)$ a global phase factor that can be dropped. An initial ground state of the resonator thus evolves into the coherent state $|\psi(t)\rangle = |\beta(t)e^{-i\omega t}\rangle$

in the presence of the gravitational wave. The probability of measuring the first excited state is $P_{0\to1}(\beta(t)) = |\langle 1|\beta(t)e^{-i\omega t}\rangle|^2 = e^{-|\beta(t)|^2}|\beta(t)|^2$. For incoherent mixtures of waves close to the mechanical resonance, in the long-time limit and for small amplitudes this yields exactly the rate (5). However, the above probability is fully general and allows for the study of single events. It is maximized for $|\beta|_{\max} = 1$. For this value of displacement the maximal probability to detect a single phonon is reached, $P_{\max} = 1/e \approx 0.37$. Larger displacements will mostly excite higher levels in the resonator. Defining $\chi(h,\omega,t) = |\int_0^t ds\,\ddot{h}(s)e^{i\omega s}|$ and again expressing the length in terms of the speed of sound $v_s = L\omega/\pi$ we get the requirement on the detector mass to obtain the maximal transition probability in the presence of $h(t)$:

$$M = \frac{\pi^2 \hbar \omega^3}{v_s^2 \chi(h,\omega,t)^2}. \tag{6}$$

This general equation provides the optimal detector mass to maximize the probability of a single gravito-phononic excitation, which quantum mechanically arises due to the absorption of a single graviton. A higher mass would increase the strain sensitivity but would result in excitations distributed among predominantly higher energy levels in the resonator. The derived result can now be applied to both continuous and transient GWs, such as from compact binary mergers which are also detected by LIGO. For the system parameters that satisfy the above equation, single gravitons are absorbed by a ground-state cooled bar resonator, which is detected by continuously monitoring the first excited level (discussed below and in the "Methods" section). The resulting excitation can be cross-correlated with a classical LIGO detection to ensure that it stems indeed from the absorption of a single graviton from the gravitational wave and confirm the corresponding frequency to verify $E = \hbar \nu$. It thus amounts to a multi-messenger approach for detecting single gravitons, overcoming the lack of control over gravitational wave sources (in contrast to the electromagnetic photo-electric analog).

   To estimate the required system parameters for single graviton detection, we take data from previously detected compact binary mergers at LIGO[33] with varying chirp masses $M_c$ and gravitational wave frequency chirp[27]. The best case is obtained for the neutron

star–neutron star (NS) merger GW170817[32], for which a slow chirp of the GW frequency through the resonance allows for a simple analytic approximation (see the "Methods" section): $M \approx \frac{24\pi^2}{5} \frac{\hbar}{h_0^2 v_s^2} \left(\frac{GM_c}{2c^3}\right)^{5/3} \omega^{8/3}$, where $h_0$ is the GW amplitude. For a beryllium resonator at frequency $\omega = 2\pi \times 100$ Hz and the GW amplitude at resonance $h_0 \sim 2 \times 10^{-22}$ this yields $M \sim 15$ kg.

Thus a resonator with such a modest mass would be able to detect a single graviton from an event such as GW170817. Other GW sources are analyzed using numerical integration of available LIGO data and are also well captured by the analytic stationary phase approximation of $\chi$. Some representative examples are summarized in Table 1. In terms of material, a higher speed of sound is desirable, but other features can make other materials more favorable, such as the tunability of the resonance frequency in superfluid Helium detectors[34,35]. The required device parameters are challenging and, in some cases, unfeasible, but for some sources, such as GW170817 they are remarkably attainable. We note that LIGO detections of such NS mergers are frequently expected with continued upgrades.

Apart from detection in the LIGO band, where low frequencies and transient sources pose the main challenges, one can also consider searches at higher frequencies and from speculative GW sources. Continuous GWs are expected in the kHz range from millisecond pulsars with asymmetric mass distributions. No such waves have yet been confirmed, and thus the strain is unknown, but observations are ongoing[36], and it was shown that resonant mass detectors, as we consider here, are well suited for such searches[34]. For this case we can estimate the graviton absorption using the stimulated rate (5), since the detector would be on resonance for a long duration. Given otherwise fixed system parameters, the rate and excitation probability increases with the resonant frequency. We require the rate to yield a single event during detector operation. The required parameters for the detection of GWs from some pulsars are summarized in Table 1. Gravitational waves have also been predicted from frequencies of up to $10^{10}$ Hz from a range of speculative sources[37], with several proposed and active experiments dedicated to their detection[38–42]. In particular, resonant mass detectors considered in this work are in use for high-frequency gravitational wave detection[39,40], and we give the single-graviton detection requirements for some examples in Table 1. Such sources, while hypothetical, could provide a near-term goal for single graviton searches, as the system parameters could be attainable with current technology for potential rare events. For example, at a gravitational wave frequency of 5.5 MHz with strain amplitude $h_0 \sim 10^{-16}$, a mass as low as $M \sim 10$ g could be used to achieve the absorption of a single graviton with our protocol. These parameters are close to currently active resonant-mass antennas that reported a rare event at these frequencies[40].

## Experimental feasibility

For our protocol the bar detector mode needs to be initiated in a single energy eigenstate, thus ground state cooling is required. The number state lifetime at $k_B T \gg \hbar\omega$ is $\hbar Q/k_B T$, where $Q$ is the acoustic quality factor. However, thermal fluctuations have to be avoided as they can mimic a stimulated process. Thermal fluctuations at temperature $T$ in the resonator yield a rate of excitation $\gamma_{th} = \omega \bar{n} Q^{-1}$ where $\bar{n}^{-1} = \exp(\hbar\omega/k_B T) - 1$. We set our benchmark such that the probability of thermal excitation during a full measurement window of roughly $10\times$ GW duration is lower than the stimulated absorption. Using again the event GW170817 we find $Q \sim 10^{10}$ and $T \sim 1$ mK. Such parameters are challenging to achieve, but they are within experimental reach: Bar detectors in operation achieved in some cases $T \lesssim 100$ mK and $Q \sim 10^8$ [35,39,43,44], and further improvements were envisioned for such detectors that come close to the required temperature and noise isolation requirements for our purposes. A summary of many previously achieved and proposed bar detector parameters is given in Fig. 2, with a comparison to our requirements. During the development of resonant bar detectors for gravitational wave detection, systems operating close to the quantum ground state were envisioned[44]. For example, the 1100 kg Auriga detector was cooled to occupation of 4000 phonons[45], while $Q \sim 10^{10}$ has been demonstrated in higher frequency resonant-mass detectors[39]. At lower frequencies, the Niobe bar detector has demonstrated a $Q$-factor of $Q \sim 10^8$ of a mode with a frequency of 700 Hz[46]. This is close to where gravitational waves have now been confirmed[32] and where signals from neutron-star-mergers are expected[47], which are the ideal sources for our purposes of single graviton detection, as outlined in Table 1. Comparing to demonstrated and proposed systems, improvements in $Q/T$ by at least two orders of magnitude are necessary for the single graviton detection regime we propose here. For other potential sources, the required $T$ and $Q$ values are reported in Table 1. We note that the noise estimates can be further improved taking into account correlations with LIGO detections or across several independent devices.

Above we focused on thermal noise in the resonator, but many more noise sources will be present. Noise from the electronics for

## Table 1 | Selected system parameters for various GW sources for the detection of single gravitons through stimulated absorption

| GW source | GW170817 (NS–NS merger) | GW170817 (NS–NS merger) | GW170608 (BH–BH merger) | GW150914 (BH–BH merger) | J1301+0833 (black-widow pulsar) | J1748-2446ad (fast-spinning pulsar) | A0620-00 (BH Superradiance) | Primordial (rare BH–BH merger) |
|---|---|---|---|---|---|---|---|---|
| $f = \frac{\omega}{2\pi}$ | 100 Hz | 150 Hz | 175 Hz | 200 Hz | 1085 Hz | 1433 Hz | 33 kHz | 5.5 MHz |
| $h_0(f)$ | $2 \times 10^{-22}$ | $2 \times 10^{-22}$ | $2 \times 10^{-22}$ | $10^{-21}$ | $<10^{-25}$ | $<10^{-25}$ | $3 \times 10^{-21}$ | $10^{-16}$ |
| $M_c$ | $1.19 M_\odot$ | $1.19 M_\odot$ | $7.9 M_\odot$ | $28.6 M_\odot$ | Continuous | Continuous | Continuous | $5 \times 10^{-4} M_\odot$ |
| Material | Beryllium | Aluminum | Niobium | CuAl6% | Niobium | Superfluid He-4 | Sapphire | Quartz |
| $v_0$ | 13 km/s | 5.4 km/s | 5 km/s | 4.1 km/s | 5 km/s | 238 m/s | 10 km/s | 6.3 km/s |
| $T$ | 1 mK | 1 mK | 1 mK | 1 mK | 0.1 μK | 0.1 μK | 0.6 K | 0.6 mK |
| $Q$-factor | $10^{10}$ | $10^{10}$ | $10^{10}$ | $10^{10}$ | $10^{10}$ | $10^{13}$ | $10^{10}$ | $10^{10}$ |
| $M$ | ~15 kg | ~250 kg | ~9 t | ~6 t | >52 t | >20 t | ~100 kg | ~10 g |

Here, $f$ is the resonant frequency, $h_0(f)$ the GW amplitude at that frequency, $M_c$ the chirp mass of the GW source in terms of solar mass $M_\odot$ where applicable, $v_0$ the speed of sound, $T$ the environmental temperature, $Q$ the mechanical $Q$-factor of the mode and $M$ the mass of the required resonator. Experiments in the LIGO band would focus on transient GWs (first four columns) and could correlate events to LIGO detections of black hole (BH) or neutron star (NS) mergers. Nearby NS mergers[32] with low chirp mass $M_c$ provide the best candidates, shown in the first two columns. For higher frequencies, the sources are of a speculative nature, such as continuous GWs from pulsars at kHz frequencies[34,36], and speculative sources due to new physics in the ultra-high frequency range[37]. The last column corresponds to a possible rare event of such a hypothetical source[40]. For continuous sources, the temperature is calculated which ensures the graviton absorption rate given in Eq. (5) is larger than the rate of thermal phonons for the given $Q$-factor and frequency. For transient sources, the temperature is calculated such that the thermal-phonon rate integrated over a 40 s observation-window yields an excitation probability lower than $P \approx 0.3$.

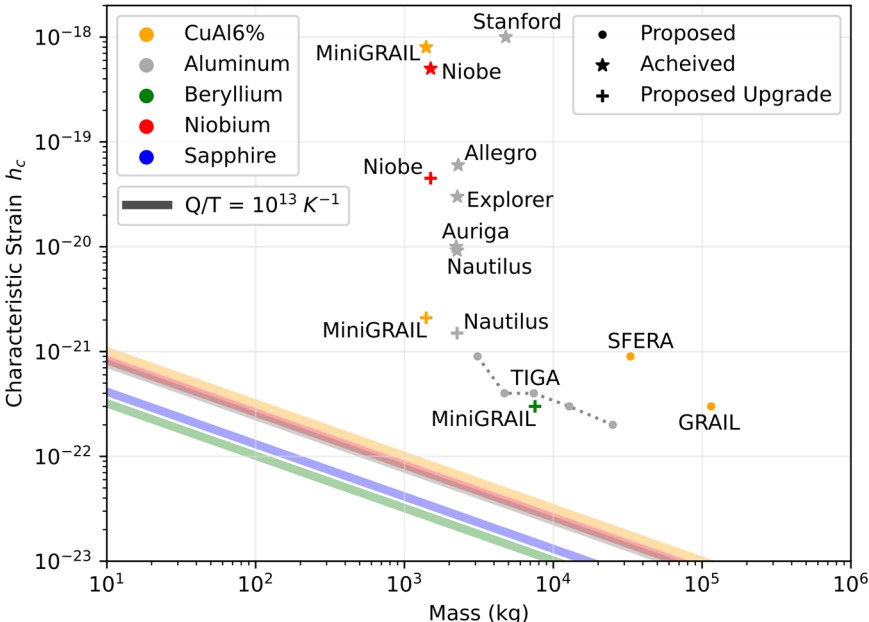

**Fig. 2 | Comparison of the effective strain sensitivity between our proposed system parameters and previously achieved and proposed bar detectors for classical gravitational wave detection.** We convert the graviton detection rate into a characteristic strain sensitivity for our proposed system (see the "Methods" section), plotted as colored lines for $Q/T = 10^{13}\,K^{-1}$ for various materials. These are compared to previously operating and proposed bar detectors of different materials and designs. The frequencies of the resonant bars are not shown, they are around 700 Hz for Niobe[46] and the proposed Grail detector[44], 842 Hz for the Stanford 1982 detector[73], around 900 Hz for Allegro, Auriga, Explorer and Nautilus[43,44,50], around 1 and 1–2 kHz for the SFERA and TIGA proposals, respectively[44], and 2.9 kHz for miniGrail[74]. Our envisioned single graviton detector would ideally operate at lower frequencies to allow for coincidence measurements with LIGO to confirm stimulated absorption from a GW.

example can be captured by an effective noise temperature. Our computed values of $T$, $Q$ dictate the tolerable values of overall noise also from other sources that can be described in terms of an effective temperature, and not just thermal fluctuations. Thus in addition to cryogenic operation, additional noise mitigation is necessary to reduce other sources, which is also one of the main challenges for classical GW detection with resonant bars. For example, a detector noise temperature for a bar detector as low as ~2 mK was demonstrated in 1995[46]. It should be noted that previously operational resonant bar detectors focused on continuous readout of transduced position, and these linear motion detectors are limited by the Standard Quantum Limit[27,44]. In contrast, a non-linear QND readout scheme that is required for our purposes is able to surpass this limit by measuring energy directly[48]. In addition to intrinsic noise of the detector, external sources have also to be mitigated. As an example, a cosmic shower directed at the center of a typical bar resonator in the energy range of ~1 GeV will have sufficient energy to excite the bar, which makes it an important noise source to be accounted for. However, this energy range is comparable to the known bounds for particle background events that must be excluded to have quantum limited sensing, as has been analyzed for classical gravitational wave detection with bar resonators[49]. Such bar detectors were equipped with cosmic ray sensors to exclude these events[50]. Therefore similar but improved strategies to eliminate cosmic noise by using better shielding or by placing the detector underground, and by filtering particle background events through coincidence measurements, would help achieve the required parameters. Overall, our graviton detection scheme benefits from decades-long development of bar detectors, but requires additional improvements as illustrated in Fig. 2 by comparing the effective strain sensitivity of our proposed detector (as computed in Methods) to previous classical bar designs, showing the required improvement in sensitivity and noise mitigation. However, we note that our detector is not optimized to be sensitive to classical GW detection, for which larger masses are favorable. Instead, we require low-mass operation to

fulfill condition (6) for optimized single graviton absorption, and operation at hectoHz is favorable for cross-correlation with LIGO.

Our proposed scheme relies on the capability to continuously monitor the energy levels of the mechanical resonator without disturbing the interaction with GWs. The key to infer a single graviton from the stimulated process is to confirm individual quantum transitions in energy, rather than measuring the average energy which would always be consistent with a classical process, as we discuss further in the Supplementary Discussion. This is a critical difference to the position measurements that have so far been considered for GW detectors. Using massive systems, the above discussed device parameters optimize the probability that only single gravitons are absorbed. In our proposed protocol the system is initially prepared in the ground state $|0\rangle$. As derived above, to first order in $h$ it evolves under the gravitational wave to state $|0\rangle + \beta|1\rangle$. The graviton is inferred when a single excitation is detected in the resonator through a number-resolving measurement. Since we do not know a priori when a GW is passing, we want to weakly and continuously monitor the excited state[51,52]. For an unsuccessful run, the ground state is re-prepared and the continuous measurement is repeated. Each run should be of duration much longer than the expected passage through resonance of a transient, chirping GW, and of maximal time $Q/(\omega\bar{n})$ for a continuous source. If a GW was present, after a characteristic time-scale $t_m$ the measurement will lead to the outcome $|1\rangle$ with probability $|\beta|^2$. The time-continuous measurement of energy is modeled in Methods, and the procedure is simulated in Figs. 3 and 4.

Both ground state cooling of massive mechanical resonators[13,53], and number-resolving phonon energy measurements[54,55] have been achieved experimentally in various devices. In particular cooling of internal phonon modes of acoustic resonators close to the ground state has been achieved[56,57], as well as the center-of-mass mode of a kg-scale LIGO mirror[14]. Furthermore, progress has been made in cooling of the bulk modes of resonant mass detectors at frequencies on the order of ~100 Hz[44,45]. The measurement of individual energy levels of

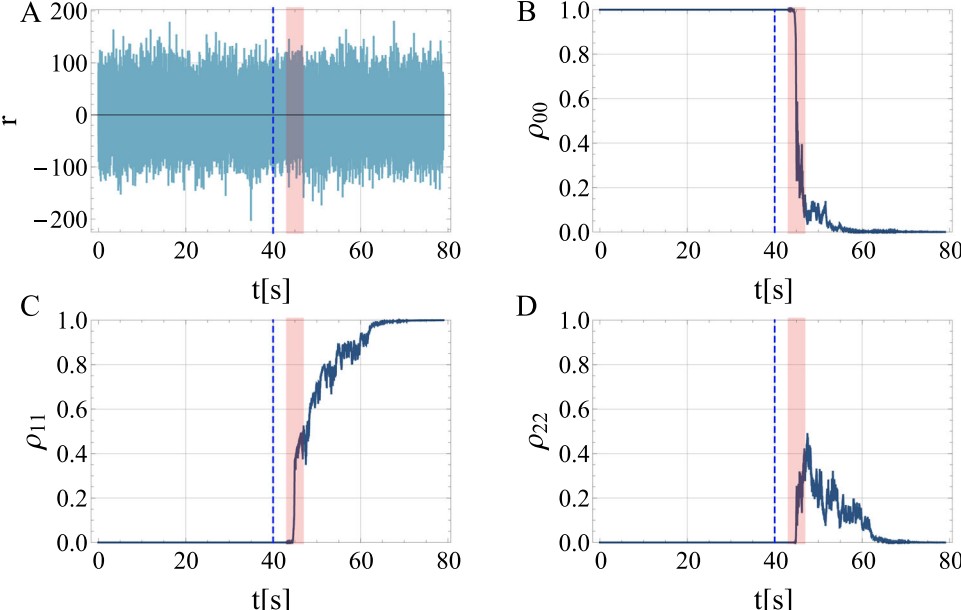

**Fig. 3 | Simulation of time-continuous and weak, number-resolving quantum measurements of an acoustic bar resonator, demonstrating absorption of a single graviton.** We show **A** the time-continuous readout signal, **B** the ground state population $\rho_{00}$ as a function of time, and **C** the population $\rho_{11}$ of the first excited state. The population of the second excited state $\rho_{22}$ is also shown (**D**). We reinitialize the detector to its quantum ground state at $t = 40\,\text{s} \lesssim Q/(\epsilon\omega\bar{n})$, indicated by the blue dashed line. In the time-window marked in red, we consider incidence of a GW with chirp mass $M_c = 1.19\,M_\odot$ and duration 4 s. The detector has mass $M = 21.73\,\text{kg}$ and frequency $\omega/2\pi = 100\,\text{Hz}$. We take $dt = 0.001\,\text{s}$ with data shown at $3dt$ intervals, $t_m = 2\,\text{s}$, and $h_0 = 2 \times 10^{-22}$. We truncate the Hilbert space dimension to 30 for the simulations. After GW incidence, in this particular run a single energy excitation is produced and confirmed.

bulk resonators, key to our proposal, has recently been achieved as well[55], albeit at microgram masses. In addition, remarkable results with continuous monitoring have been achieved in various quantum devices[58,59], including massive mechanical oscillators[60]. Nevertheless, engineering direct coupling to the energy of the acoustic mode for continuous measurements at such masses as required for our proposed experiment is the most significant challenge. This amounts to quantum sensing capabilities to discern energy levels at pico-eV resolution for kg-scale masses. While very challenging, there are several possibilities for achieving this for our purposes. One possibility is to couple electromagnetic waves to the fundamental acoustic mode through an optomechanical non-linear coupling, such as in a membrane-in-the-middle setup. Another possibility is to use surface acoustic waves, such as with an analog of Brillouin optomechanics for electromagnetic cavity frequencies—if the electromagnetic modes are on the order of MHz, phase-matching as required for Brillouin scattering can be achieved for acoustic modes on the order of hundreds of Hz. It is also of interest to investigate transducers as in conventional bar detectors. Alternatives to traditional bar resonators such as superfluid Helium resonators[34,35] may also offer new capabilities to engineer the required energy sensing in controlled environments. Thus, while the readout is a significant challenge, further improvements and symbiosis of various capabilities to prepare and probe macroscopic quantum systems offer paths for the development of a graviton detector, building on modern quantum measurement capabilities in massive resonators. Arrays of such acoustic detectors, tolerance of lower detection probability or the use of transducers to lower the mass requirements may significantly help in achieving the required energy sensing.

As discussed above, the experimental realization of our proposal requires improvement to current technology. Building on decades-long development of Weber bar detectors[27,39,44,50,61], our proposal requires such acoustic resonators to operate at the ground state and be augmented with quantum sensing of individual energy levels. We stress, that the outlined proposal has the advantage that it does not rely on quantum state preparation beyond ground state cooling, and in particular no macroscopic superpositions. Validating the quantum nature of the gravito-phononic excitations relies instead on direct energy measurements such as in ref. 55, which involves coupling a meter to the energy operator of the acoustic mode (see the "Methods" section). Such dispersive or QND couplings to energy are not restricted by the Standard Quantum Limit for position measurements, which is a challenge in transducer-based read-out systems typically used in bar detectors. Moreover, correlating detection events with LIGO if in the same frequency range can further reduce the noise constraints that have so far been limiting gravitational bar detectors[44]. In general, position measurements that have to date been the focus of GW detection would not be able to resolve individual transitions between discrete energy levels and could only provide information on the average energy transfer. It is the ability to continuously monitor and detect changes in single energy quanta that enables graviton inference through absorption from the GW.

## Interpretation

The graviton detection scheme discussed here, and its experimental requirements, are of a different nature than other proposals for table-top tests of quantum gravity. Such proposals have so far either focused on testing modified quantum dynamics due to speculative quantum gravity phenomenology[2–5,10,11], or entanglement generated by gravitating source masses in superposition[6,7]. The latter rely on the very challenging creation of macroscopic quantum superpositions. In contrast, in our work the state preparation only relies on ground state cooling, but the difficulty lies in implementing the quantum sensing scheme for massive systems. Our work also focuses on a very different regime of the interaction between matter and gravity, in contrast to other schemes: here we show that exchange of single quanta of energy with gravitational waves can be observed, while proposals to test entanglement generation through gravity focus on the expected Newtonian interaction between static masses in superposition. Arguments can be made that the latter also provide a signature of graviton

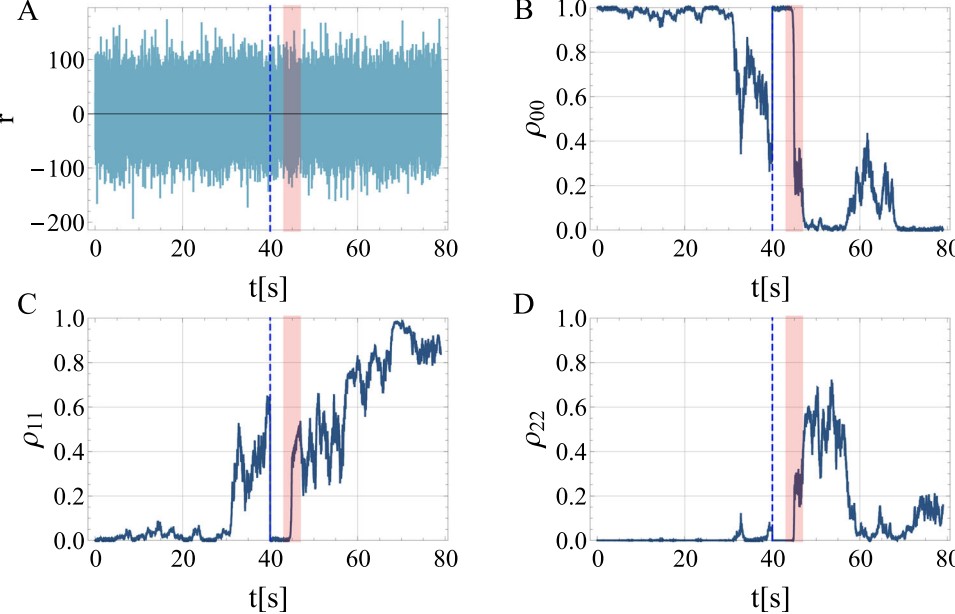

**Fig. 4 | A different realization compared to Fig. 3 where we also incorporate a random unitary displacement $D(\gamma)$ at each timestep as additional noise. Both $\text{Re}(\gamma)$ and $\text{Im}(\gamma)$ are assumed to be Gaussian random variables of variance $\kappa^2/dt$, and mean zero. We take $\kappa = 0.5 \times 10^{-4}$.** The panels again show **A** the time-continuous readout signal, **A** the ground state population $\rho_{00}$, and the population of the **C** first excited state $\rho_{11}$ and **D** the second excited state $\rho_{22}$ in the bar. After GW incidence, the excitation can be inferred from the measurement, and correlation to independent LIGO detection could confirm the GW as the source with high confidence.

exchange, but for virtual particles[62] and under additional assumptions[63]. In contrast, our work here relies on the direct on-shell exchange of gravitons. Such tests therefore test different but complementary aspects of gravity in the quantum regime.

As mentioned before, it is sufficient to use the semi-classical limit of the interaction Hamiltonian (3) to derive our results—single phonon transitions. The experiment therefore cannot be used as proof of the quantization of gravity. It does not reveal the quantum state of the graviton, but the stimulated single-graviton emission and absorption. This is directly analogous to the original photoelectric effect: Lamb and Scully showed and discussed semi-classical models for the photoelectric observations[64]. However, this semi-classical limit would require the violation of energy conservation for single discrete transitions in energy. Our focus here is thus the exchange and verification of single quanta of energy that can serve as a first evidence of quantumness[65], rather than a direct proof. The continuous monitoring of energy levels is thus again key to our proposal: In analogy with the photoelectric effect, assuming energy conservation at the level of individual transitions between field and matter, individual quantum jumps in energy are evidence of the absorption or emission of a single graviton of energy $E = \hbar\nu$. This interpretation is valid close to resonance and in the rotating-wave approximation, which is the relevant regime for our scheme. The comparison to the photo-electric effect is discussed in more detail in Supplementary Discussion. We note that the observation of the exchange of single quanta between gravitational waves and matter may also offer surprises and proceed differently from the expected behavior described here, and empirical experimental evidence is highly desirable even if the linearized limit of the interaction is well understood theoretically. The realization of our proposed experiment would effectively constitute the gravito-phononic analogy with the observation of the photoelectric effect, historically the first evidence of the quantization of light.

## Discussion

In summary, we derived from first principles the absorption and emission rates for single gravitons in macroscopic mechanical resonators operating in the quantum regime. Despite interacting with essentially classical waves with $\geq 10^{36}$ gravitons, the interaction is weak enough such that only single quanta are exchanged on the relevant time scales. This requires challenging but attainable parameters. We found that detection of stimulated absorption of single gravitons from known sources of gravitational waves are within reach of near future experiments, such as with ground-state-cooled kg-scale bar resonators with continuous quantum sensing of its energy. Correlating measurements with classical LIGO detection events can confirm GWs as the source, and that discrete energy $E = \hbar\nu$ is exchanged. The scheme could also operate at higher frequencies, but with uncertainty about possible sources. In analogy with the electromagnetic photoelectric effect, such detections would provide the first sign of gravitons, giving the most compelling experimental indication to date for quantum gravity.

## Methods

### Microscopic derivation of collective interaction

Here, we present a fully quantum mechanical treatment of the interaction between a gravitational wave and a solid-bar resonator from the perspective of the dynamics of individual atoms in the solid. The analysis closely follows the approach in ref. [66], however, some of the key differences are highlighted. We restrict the analysis to one dimension, which spans across the length of the solid-bar resonator. In this simplified picture, the solid-bar resonator can be thought of as a collection of $N+1$ atoms ($N$ odd) with nearest-neighbor interactions. We treat the atoms as identical having mass $m$ each, so the total mass of the solid-bar resonator is $(N+1)m$. Given this, as illustrated in Fig. 5, the vibrations of each of the atoms can be modeled as simple harmonic oscillations about their mean position $x_n = an/2$, where $a$ is the lattice spacing, and $n$ is an odd number such that $-N < n < N$. As an example, for $N = 5$, we have $N+1 = 6$ atoms, whose mean positions with respect to the center of mass (at $x = 0$) of the solid-bar resonator are respectively at $x = \pm a/2, \pm 3a/2, \pm 5a/2$.

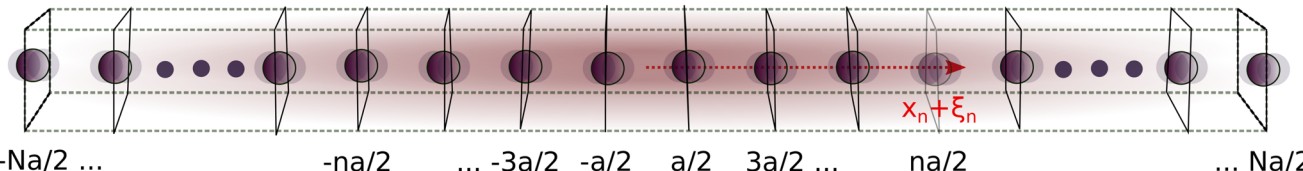

-Na/2 ...                    -na/2        ... -3a/2  -a/2   a/2  3a/2 ...     na/2                    ... Na/2

**Fig. 5 | A solid-bar resonator modeled as a collection of atoms vibrating about their respective mean positions $x_n = na/2$.** Here $a$ is the lattice spacing, and $\xi_n$ is the displacement of an atom in simple harmonic motion about its mean position $x_n$.

The equation of motion for the $n$th atom is

$$m\ddot{\xi}_n + m\omega_D^2(2\xi_n - \xi_{n-2} - \xi_{n+2}) = 0. \tag{7}$$

By making a comparison to a chain of coupled masses which are identical with mass density $\rho = M/L \approx m/a$, the tension per unit length $m\omega_D^2$ is related to the speed of sound $v_s$ in the continuum limit by $m\omega_D^2 \approx \rho v_s^2/a = mv_s^2/a^2$, where $v_s$ is the speed of sound. Hence the frequencies of oscillations $\omega_D$ with respect to the neighboring atoms is approximately the Debye frequency. Above, note that $\xi_n$ is the displacement of the atom around its mean position $x_n$. In other words, the position $x$ of the $n$th atom with respect to the center of mass of the solid-bar resonator (at $x = 0$), when the atom's simple harmonic motion is also accounted for, is given by $x \approx x_n + \xi_n$.

The general solution to Eq. (7) is given by

$$\xi_n(t) = e^{-i\omega t}(Ae^{ikna/2} + Be^{-ikna/2}) + \text{H.c.}, \tag{8}$$

where the dispersion relation is given by, $\omega^2 = 2\omega_D^2[1 - \cos(ka)]$. Assuming no elastic energy is flowing out of the bar, we have the boundary conditions that

$$\left.\frac{d\xi_n}{dn}\right|_{n=\pm N} = 0 \tag{9}$$

This suggests that $k$ is discrete, and the solution $\xi_n(t)$ can be written as a discrete sum over different modes each satisfying the boundary conditions

$$\xi_n(t) = \sum_{l=0,2..}^{N-1} \chi_l(t)\cos[l\pi n/(2N)] + \sum_{l=1,3..}^{N} \chi_l(t)\sin[l\pi n/(2N)]. \tag{10}$$

However, the completeness relation for sine and cosine over a discrete sum is given by (recall that $n$ is odd)[66]

$$\sum_{n=-N}^{N} \cos\left[\frac{l\pi n}{2(N+1)}\right]\cos\left[\frac{l'\pi n}{2(N+1)}\right] = \frac{N+1}{2}\delta_{ll'},$$

$$\sum_{n=-N}^{N} \sin\left[\frac{l\pi n}{2(N+1)}\right]\sin\left[\frac{l'\pi n}{2(N+1)}\right] = \frac{N+1}{2}\delta_{ll'}, \tag{11}$$

$$\sum_{n=-N}^{N} \cos\left[\frac{l\pi n}{2(N+1)}\right]\sin\left[\frac{l'\pi n}{2(N+1)}\right] = 0.$$

So a small correction to the solutions to account for discreteness of the system is necessary, which modifies the solutions to,

$$\xi_n(t) = \sum_{l=0,2..}^{N-1} \chi_l(t)\cos\left[\frac{l\pi n}{2(N+1)}\right] + \sum_{l=1,3..}^{N} \chi_l(t)\sin\left[\frac{l\pi n}{2(N+1)}\right]. \tag{12}$$

As noted in ref. 66, the change is negligible in the large $N$ limit, when $N \sim N+1$. Also, note that the equation above differs from that in ref. 66 by a scaling factor of $\sqrt{2}$ for $l \neq 0$ modes.

Recall that $\xi_n(t)$ are required to satisfy Eq. (7). By using the orthonormal conditions above, we find that $\chi_l$ satisfies the equation of

motion for another set of simple Harmonic oscillators, given by[66]

$$\ddot{\chi}_l + \omega_l^2\chi_l = 0, \tag{13}$$

where $\omega_l^2 = 2\omega_D^2(1 - \cos[l\pi/(N+1)])$. To compute the mass-energy of each of these normal modes, one can compare the energy density

$$\begin{aligned} E &= \frac{m}{2}\sum_{n=-N}^{N}\dot{\xi}_n^2 + \frac{m\omega_D^2}{2}\sum_{n=-N}^{N-2}(\xi_{n+2} - \xi_n)^2 \\ &= \frac{M}{4}\sum_{l=0}^{N}\dot{\chi}_l^2 + \frac{M}{4}\sum_{l=0}^{N}\omega_l^2\chi_l^2. \end{aligned} \tag{14}$$

This shows that each normal mode can be treated as an oscillator of mass $M/2$. In comparison, note that in ref. 66 each $l$ mode is interpreted as of mass $M$. However, we believe this was due to an extra scaling of $\sqrt{2}$ for $l \neq 0$ modes used in ref. 66. For identical atoms, we can also confirm that the mass of each mode $l$ has to be $M/2$ by taking the continuum limit where each of the $\chi_l$ are acoustic modes. In the continuum limit, acoustic modes $\chi_l(x)$ satisfy[27]

$$\int_{-L/2}^{L/2} dx\chi_l(x)^2 = L/2. \tag{15}$$

Assuming a uniform mass distribution $\rho(x) = M/L$ (corresponding to identical atoms in the discrete scenario) the reduced mass is given by

$$\mu_l = \int_{-L/2}^{L/2} dx\rho(x)\chi_l(x)^2 = \frac{M}{L}\int_{-L/2}^{L/2} dx\chi_l(x)^2 = M/2. \tag{16}$$

## The interaction Hamiltonian

We now proceed to derive the interaction Hamiltonian for the interaction with a Gravitational wave. The force on each of the atoms can be written as the mass of the atom times the gradient of the gravitational potential, expanded about the center of mass of the bar resonator:

$$\begin{aligned} f(x) &= -m\nabla\phi = -m\nabla\left(\frac{1}{2}\frac{\partial^2\phi}{\partial x^2}x^2 + ...\right) \\ &= m\frac{\ddot{h}_{xx}}{4}\nabla(x^2) = m\frac{\ddot{h}_{xx}}{2}x. \end{aligned} \tag{17}$$

Above, we have replaced $\frac{\partial^2\phi}{\partial x^2} = -\frac{1}{2}\ddot{h}_{xx}$ using the weak-field limit and have assumed that the length of the bar resonator is much smaller than the wavelength of the gravitational wave such that $\ddot{h}_{xx}$ is effectively constant across the bar. Now recall that particle coordinates are given by discrete values of $x = x_n + \xi_n$, accounting for the oscillation of each of the atoms about their mean positions, $x_n = na/2$. This yields the force on individual atoms as

$$f_n = m\frac{\ddot{h}_{xx}}{2}(x_n + \xi_n). \tag{18}$$

The force computed above differs from that in ref. 66, where the atoms' positions were approximated to be $x = x_n$. This is a good approximation and derives the leading contribution to interaction

with a gravitational wave in the subsequent paragraphs. However, we note that it ignored the displacement of each of the atoms $\xi_n$ about their center of mass $x_n$. Accounting for this small displacement gives us an insightful, sub-leading contribution to the interaction with a gravitational wave, discussed later in this section. Our results can also be obtained using directly Eq. (1) in the main text: choosing the local inertial frame, the metric is given by $h_{00} = -\frac{1}{2}\ddot{h}_{xx}(x_n + \xi_n)^2$.

Using Eq. (18), we obtain the total interaction energy by summing over all the contributions of atoms in the solid-bar resonator, as

$$H_I = -m\frac{\ddot{h}_{xx}}{2}\sum_{n=-N}^{N}(x_n\xi_n + \xi_n^2/2). \qquad (19)$$

Completing the first summation over $n$ with $x_n = na/2$, we get,

$$-m\frac{\ddot{h}_{xx}}{2}\sum_{n=-N}^{N}x_n\xi_n \approx -\frac{ML\ddot{h}_{xx}}{\pi^2}\sum_{l=1,3...}^{N}(-1)^{\frac{l-1}{2}}\frac{1}{l^2}\chi_l(t). \qquad (20)$$

Here, the sum over all atoms results in an enhanced effect that builds up across the length of the resonator, scaling with mass and length. The second term similarly gives (using the completeness relations)

$$-m\frac{\ddot{h}_{xx}}{4}\sum_{n=-N}^{N}\xi_n^2 = -\frac{M\ddot{h}_{xx}}{8}\sum_{l=0}^{N}\chi_l^2 \qquad (21)$$

Therefore upon quantization, the total interaction Hamiltonian is

$$\hat{H}_I = \sum_{l=0}^{N}\hat{H}_I^l = -\frac{ML\ddot{h}_{xx}}{\pi^2}\sum_{l=1,3...}^{N}(-1)^{\frac{l-1}{2}}\frac{1}{l^2}\hat{\chi}_l - \frac{M\ddot{h}_{xx}}{8}\sum_{l=0}^{N}\hat{\chi}_l^2. \qquad (22)$$

From Eq. (14) we know that each of the $l$ modes have mass $M/2$. Therefore, the interaction Hamiltonian for each of modes $l$ (given $l$ is odd) has the following form in terms of the annihilation operator for the mode $\hat{b}_l$

$$\hat{H}_I^{l,\text{odd}} = -\frac{ML\ddot{h}_{xx}}{\pi^2}(-1)^{\frac{l-1}{2}}\frac{1}{l^2}\sqrt{\frac{\hbar}{M\omega_l}}(\hat{b}_l + \hat{b}_l^\dagger) - \frac{\ddot{h}_{xx}}{8}\frac{\hbar}{\omega_l}(\hat{b}_l + \hat{b}_l^\dagger)^2. \qquad (23)$$

For even $l$, we have

$$\hat{H}_I^{l,\text{even}} = -\frac{\ddot{h}_{xx}}{8}\frac{\hbar}{\omega_l}(\hat{b}_l + \hat{b}_l^\dagger)^2. \qquad (24)$$

It is of pedagogical interest to compare the interaction Hamiltonians in Eqs. (22) and (23) above to the quadrupole interaction Hamiltonian derived in ref. 22. The equivalent (to the interaction Hamiltonian for a Hydrogen atom with a gravitational wave, in the local inertial frame) would be the second term in Eqs. (22) and (23). Although present, we note that such an interaction is only a sub-leading contribution for solid-bar resonators. This term represents a direct quadrupole interaction for quantum oscillators (each with quadrature $\chi_l$, mass $M/2$) with a gravitational wave, in the local inertial frame. It is also evident from (the second term of) Eq. (19) that such a direct quadrupole interaction originates from the quadrupole coupling of a gravitational wave to each individual atom of the solid-bar, having quadrupole moment $Q_{\xi\xi} = \xi^2$. We also see that even modes ($l$ even) only experience this direct quadrupole interaction as shown in Eq. (24).

However, what is unique and interesting here for a solid-bar resonator is that there is an additional macroscopic effective interaction with a gravitational wave experienced by odd $l$ modes, which, in fact, is the leading contribution. This dominant contribution is the first term in Eqs. (22) and (23), and it emerges from the interaction of odd acoustic ($l$ odd) modes with a gravitational wave through the gradient of the quadrupole moment of the solid bar, with the center of mass of

the bar-resonator as the reference point. It is evident from Eq. (20) that this is an integrated effect that builds up across the length of the solid-bar resonator and, therefore, represents an advantage for detecting gravitons using bar resonators as opposed to individual atoms. The ability to address individual acoustic modes in an experiment further enhances the feasibility of experimental tests in this regime.

**Stimulated absorption rate in the quantum picture**

For computing the quantum interaction, we approximate that the interaction Hamiltonian only includes the first term in Eq. (23). Quantizing the gravitational field perturbation as above, and substituting it into the interaction Hamiltonian, we obtain for a plane wave

$$\hat{H}_{\text{int}} = \hbar\sqrt{(-1)^{l-1}\frac{8\pi GM\nu^3}{\omega_l c^2 V}}\frac{L}{\pi^2 l^2}\left(\hat{b}_l + \hat{b}_l^\dagger\right)\left(\hat{a}e^{-i\nu t} + \hat{a}^\dagger e^{i\nu t}\right), \qquad (25)$$

which is valid under the dipole approximation, as well as the single mode approximation—the mechanical resonator only interacts with a single mode of the gravitational field, namely the mode on resonance with the resonator mode, with annihilation (creation) operator $\hat{a}$ ($\hat{a}^\dagger$).

We now calculate the transition rate for which the initial state is $|i\rangle = |\alpha\rangle|0\rangle$ and the final state is $|f\rangle = |\alpha\rangle|1\rangle$, where $|\alpha\rangle$ corresponds to a coherent state of the gravitational field. This transition corresponds to the mechanical resonator absorbing a single graviton. Since the gravitational field is approximately in a coherent state, such an absorption does not change its state (this is only a convenient approximation, as a coherent state assumes infinitely many number states). We compute the stimulated rate from Fermi's golden rule

$$\Gamma_{\text{stim}} = \frac{2\pi}{\hbar^2}|\langle\alpha|\langle 1|\hat{H}_{\text{int}}|\alpha\rangle|0\rangle|^2 D(\omega), \qquad (26)$$

where $D(\omega)$ is the graviton density of states. For the above Hamiltonian in the interaction picture and on resonance in the rotating-wave approximation, this gives the following absorption rate:

$$\Gamma_{\text{stim}} = \frac{|\alpha|^2}{l^4}\frac{8GML^2\omega_l^4}{\pi^4 c^5}. \qquad (27)$$

The number of gravitons in a gravitational wave signal for a given strain amplitude $h_0$ is[22,23]

$$N = \frac{h_0^2 c^5}{32\pi G\hbar\nu^2}. \qquad (28)$$

This result is obtained by dividing the wave's energy density $\rho_E = \frac{c^2}{32\pi G}h_0^2\nu^2$ by the energy density of a single graviton with $E = \hbar\nu$ in a cubic box of size $c/\nu$. As an example, taking a strain amplitude of $h_0 = 10^{-21}$, and a gravitational wave frequency of $\nu/2\pi = 150$ Hz, one gets $N \approx 4 \times 10^{36}$.

In order to convert the absorption rate in Eq. (27) into the transition rate due to interaction with a classical gravitational field, we make the replacement $|\alpha|^2 \to N$ in Eq. (27)

$$\Gamma_{\text{stim}} = \frac{1}{l^4}\frac{ML^2\omega_l^2 h_0^2}{4\pi^5\hbar}, \qquad (29)$$

fully consistent with the result from treating the gravitational field classically in the main article. Although Eq. (29) can be explained with the gravitational wave treated as a classical field, we have shown that this absorption rate can be derived from a single graviton absorption process. In this way, the transition of the mechanical resonator from the ground state to the first excited state $|n=0\rangle \to |n=1\rangle$ can be explained as the absorption of a single graviton from a coherent state

of the gravitational field $|\alpha\rangle$, with a macroscopic number of gravitons $N = |\alpha|^2$.

It is important to note that the interaction Hamiltonian in the quantum picture conserves energy, for transitions between eigenstates within the rotating-wave-approximation. Each transition corresponds to the exchange of a $\hbar\omega$ packet of energy between field and matter, through the interaction Hamiltonian. This feature, however, is not directly discernible on the state of the field, which is approximated as a coherent state (with summation of number states to infinity) and is thus also not present in the semi-classical model.

## Exact dynamics

The semi-classical Hamiltonian for a single-mode gravitational wave $h(t)$ and a single mode of the resonator is described by

$$\hat{H} = \hbar\omega\hat{b}^\dagger\hat{b} + \frac{1}{\pi^2}L\sqrt{\frac{M\hbar}{\omega}}\ddot{h}(t)(\hat{b} + \hat{b}^\dagger) \tag{30}$$

where $\hat{b}$ acts on the resonator mode of interest. In the interaction picture, the evolution is thus governed by the operator

$$\hat{U}_{\text{int}} = \hat{T}e^{-i\int_0^t ds\left(g(s)\hat{b}(s) + g^*(s)\hat{b}^\dagger(s)\right)}. \tag{31}$$

with $\hat{T}$ the time-ordering operator, $\hat{b}(t) = \hat{b}e^{-i\omega t}$ and

$$g(t) = \frac{1}{\pi^2}L\sqrt{\frac{M}{\hbar\omega}}\ddot{h}(t). \tag{32}$$

The dynamics can be solved exactly, either using a Lie-algebra method[67], or alternatively, a time-ordered unitary operator $\hat{U} = \hat{T}e^{\int_0^t ds\hat{A}(s)}$ can also be written in terms of the Magnus expansion as

$$\hat{U} = e^{\Omega(t)} \tag{33}$$

where

$$\Omega(t) = \int_0^t dt_1\hat{A}(t_1) + \frac{1}{2}\int_0^t dt_1\int_0^{t_1}dt_2[\hat{A}(t_1),\hat{A}(t_2)] + \ldots \tag{34}$$

In our case we have $\hat{A}(t) = -i\left(g(t)\hat{b}(t) + g^*(t)\hat{b}^\dagger(t)\right)$, $[\hat{A}(t_1),\hat{A}(t_2)] = -(g(t_1)g^*(t_2)e^{-i\omega(t_1-t_2)} - g(t_2)g^*(t_1)e^{+i\omega(t_1-t_2)})$ and all higher nested commutators vanish such that the above Magnus expansion ends at the second term. This results in the interaction-picture unitary evolution corresponding to Hamiltonian (30):

$$\hat{U}_{\text{int}} = e^{-i\int_0^t ds\left(g(s)\hat{b}(s) + g^*(s)\hat{b}^\dagger(s)\right)}e^{-i\varphi} \tag{35}$$

where $\varphi = \int_0^t dt_1\int_0^{t_1}dt_1\text{Im}[g(t_1)g^*(t_2)e^{-i\omega(t_1-t_2)}]$. The full evolution is thus

$$\hat{U} = e^{-i\varphi}e^{-i\omega t\hat{b}^\dagger\hat{b}}\hat{D}(\beta), \tag{36}$$

where $\hat{D}(\beta) = e^{\beta\hat{b}^\dagger - \beta^*\hat{b}}$ is the displacement operator with strength

$$\beta = -i\int_0^t ds\, g^*(s)e^{i\omega s} \tag{37}$$

and $g(s)$ in Eq. (32).

An initial vacuum state $|0\rangle$ evolving under Hamiltonian (30) thus becomes

$$e^{-i\varphi}|\beta e^{-i\omega t}\rangle \tag{38}$$

Plugging in the physical parameters from Eq. (32) we get

$$|\beta| = \frac{L}{\pi^2}\sqrt{\frac{M}{\omega\hbar}}\chi(h,\omega,t) \tag{39}$$

with

$$\chi(h,\omega,t) = |\int_0^t ds\,\ddot{h}(s)e^{i\omega s}|. \tag{40}$$

The results show that the interaction with a coherent gravitational wave produces coherent states of the resonator. This result is directly analogous to the quantum optical case where a semiclassical interaction between current and matter produces coherent states, as first considered by Glauber[68]. The results show that the interaction with a coherent gravitational wave produces coherent states of the resonator. This result is directly analogous to the quantum optical case where a semiclassical interaction between current and matter produces coherent states, as first considered by Glauber[68]. The parameter $|\beta|$ in Eq. (39) is central to estimating the transition probability from the ground to the excited state, and it depends on the physical parameters of the detector $L$, $M$, and $\omega$ on the one hand, and on the properties of the passing gravitational wave through $\chi(h, \omega, t)$ in Eq. (40) on the other. Thus optimizing the detector for a single transition requires knowledge of the GW profile, which can be obtained from independent LIGO detections. We note, however, that also if not optimal, there is a finite probability of a single transition as given by the Poisson number distribution $P_n = e^{-|\beta|^2}|\beta|^{2n}/n!$.

## Resonator mass estimate for compact binary mergers

The above results show that a resonance is being built up between the gravitational wave and the acoustic mode, captured by $\chi(h, \omega, t)$. The function exhibits a sharp resonance around the resonator frequency $\nu = \omega$. This resonance becomes more pronounced as the integration time $t$ is increased. For a single monochromatic wave $h(t) = h_0\sin(\nu t)$ we have $\chi \sim \frac{1}{2}h_0\nu^2 t\,\text{sinc}(\frac{t\Delta}{2})$ with $\Delta = |\nu-\omega|$, using the rotating-wave approximation for $\Delta \ll \omega$, $\nu$. For a mixture of plane waves around a small resonance window this results in the golden rate in the limit $t\Delta \to \infty$, which we also used in the main text.

For transient sources, the above calculation for $\chi$ breaks down as the long-time limit is not valid. Instead one can approximate the solution with the stationary phase method. The idea is to first expand $\chi(h, \omega, t)$ using integration by parts, and evaluate the finite Fourier transform of $h(s)$ that remains using the stationary phase method. Here we Taylor expand the phase $\Phi(s)$ of $h(s)e^{i\omega s}$ (after neglecting rapidly oscillating terms) to the second order around a stationary point $s = s^*$ where $\Phi'(s)|_{s=s^*} = 0$. We then evaluate the Gaussian integral, which gives an approximate solution for $\chi(h, \omega, t)$ in closed form, providing a better estimate for $\chi(h, \omega, t)$ for transient sources.

We also present a simple analytic approximation that holds for intermediate times. For binary inspirals, to the lowest order, the emitted GW frequencies follow the equation[27] $\dot{f} = k_f f^{11/3}$, and thus with $\nu = 2\pi f$:

$$\nu(t) = \left(\frac{1}{\nu_0^{8/3}} - \frac{8}{3}kt\right)^{-3/8}, \tag{41}$$

where

$$k = \frac{k_f}{(2\pi)^{8/3}} = \frac{96}{5}\pi\left(\frac{\pi GM_c}{c^3}\right)^{5/3}\frac{1}{(2\pi)^{8/3}} = \frac{48}{5}\left(\frac{GM_c}{2c^3}\right)^{5/3} \tag{42}$$

and $M_c = (m_1 m_2)^{3/5}/(m_1 + m_2)^{1/5}$ is the effective chirp mass of a binary system with masses $m_1$ and $m_2$. The frequency of the incoming GW thus chirps, causing a transition through the resonance. For a slow transition, we can find an analytical expression for $\chi$ as follows.

We estimate the time $\tau$ that a GW has a frequency that stays within the resonance window $[\omega - \Delta\omega, \ \omega + \Delta\omega]$: $\tau = t(\nu = \omega + \Delta\omega) - t(\nu = \omega - \Delta\omega)$. This yields

$$\tau = \frac{2\Delta\omega}{k\,\omega^{11/3}}. \tag{43}$$

To estimate the resonance window we assume that the GW frequency is roughly constant during the transition through the resonance. We solve $\chi$ for the monochromatic case $h(t) = h_0 \sin(\nu t)$ such that

$$\chi = \left| \int_0^t ds\, e^{i\omega s} h_0 \nu^2 \sin(\nu s) \right| \approx h_0 \nu^2 \frac{t}{2} \mathrm{sinc}\left(\frac{\delta\,t}{2}\right). \tag{44}$$

Here we assume $\omega + \nu \gg \omega - \nu = \delta$ for the frequency range of interest. The sinc-function has its first zero at $\frac{\delta T}{2} = \pm\pi$. The FWHM for $\mathrm{sinc}(x)$ is at $x \approx 1.895 \approx 2$. Thus the frequency bandwidth is approximately

$$2\Delta\omega = \frac{8}{T}. \tag{45}$$

This is the frequency window in which the resonator interacts with the GW within a time-scale $T$. Setting this time-scale to $T = \tau$ in Eq. (43) we obtain

$$\tau = 2\sqrt{\frac{2}{k}}\omega^{-11/6}. \tag{46}$$

This is approximately the time for the GW to pass through the resonance.

We now truncate the integral in $\chi$ to this time and approximate the GW frequency by the resonant frequency during this time:

$$\begin{aligned}\chi &\approx h_0 \omega^2 \left| \int_0^\tau ds\, e^{i\omega s} \sin(\omega s) \right| \\ &= h_0 \frac{\omega}{4}\sqrt{2 + 4\omega^2\tau^2 - 2\cos(2\omega\tau) - 4\omega\tau\sin(2\omega\tau)}.\end{aligned} \tag{47}$$

For the case $\omega\tau \gg 1$ this greatly simplifies to

$$\chi(\tau) \approx h_0 \frac{\omega^2 \tau}{2}. \tag{48}$$

Using the above estimate for $\tau$, Eq. (46), we thus get

$$\chi \approx h_0 \sqrt{\frac{2}{k}}\omega^{1/6} = h_0 \sqrt{\frac{5}{24}}\left(\frac{2c^3}{GM_c}\right)^{5/6}\omega^{1/6}. \tag{49}$$

This is our analytic estimate for $\chi$, which holds for chirping GWs from binary sources that pass sufficiently slow through the resonance. The corresponding optimal mass, as discussed in the main text, is thus

$$M = \frac{\pi^2 \hbar \omega^3}{v_s^2 \chi^2} \approx \frac{\pi^2 \hbar k}{2 v_s^2 h_0^2}\omega^{8/3} = \frac{24\pi^2}{5}\frac{\hbar}{h_0^2 v_s^2}\left(\frac{GM_c}{2c^3}\right)^{5/3}\omega^{8/3}. \tag{50}$$

For the NS–NS merger GW170917[32] this gives excellent agreement with the stationary-phase method mentioned above. For other sources that have a faster chirp, the analytic estimate is less precise, but still offers a good approximation. Available LIGO data from currently detected sources[33] provides independent numerical estimates for $\chi$.

## Characteristic strain sensitivity to monochromatic waves

Here we compute the characteristic strain sensitivity of our detector to monochromatic sources. To this end, we consider a periodic gravitational wave of the form $h(t) = h_0 \sin \nu t$ that interacts with our detector. At resonance ($\nu = \omega$, the resonant frequency of the detector), it follows from Eq. (48) that the probability of observing a single excitation in the bar detector initialized in its quantum ground state, within duration $t$ is given by

$$P(t) = \frac{L^2}{\pi^4}\frac{M}{\omega\hbar}|\chi(t)|^2 = \frac{h_0^2 \omega t^2 M v_s^2}{4\pi^2 \hbar}. \tag{51}$$

Hence a strictly monochromatic wave induces excitations at the following rate:

$$\Gamma_{\mathrm{mc.}} = \frac{dP(t)}{dt} = \frac{h_0^2 \omega t M v_s^2}{2\pi^2 \hbar} = \frac{h_0^2 N_c M v_s^2}{\pi\hbar}, \tag{52}$$

where $N_c = \omega t/(2\pi)$ is the number of cycles of the gravitational wave that interact with the detector. By comparing this rate to the competing rate of thermal excitations $\gamma_{\mathrm{th}} = \omega\bar{n}Q^{-1}$, we can identify the corresponding minimum detectable strain amplitude

$$h_0 = \sqrt{\frac{\pi k_B T}{M v_s^2 Q N_c}}, \tag{53}$$

which agrees with the sensitivity requirements for detecting strictly monochromatic classical gravitational waves with a Weber bar detector[69,70].

To define the characteristic strain amplitude for our detector, it is desirable to relax the "strictly monochromatic" constraint a bit and consider the more realistic case of a wavepacket in the frequency domain around the central resonant frequency $\omega$. For an off-resonant wave from this wavepacket having frequency $\nu$, we can approximate the transition probability to the first excited state using Eq. (44) as

$$P(\nu,\omega,t) \approx |\beta(\nu,\omega,t)|^2 \approx \frac{\left(h_0^2 \omega^3 M L^2\right)\sin^2\left[\frac{1}{2}t(\nu - \omega)\right]}{\hbar(\nu - \omega)^2 \pi^4}. \tag{54}$$

Assuming contributions from different frequencies do not interfere (this can be understood as resulting from summing over GWs of uncorrelated phases), we can compute the total excitation probability as the marginal,

$$\begin{aligned}P(\omega,t) &\approx \sum_\nu |\beta(\nu,\omega,t)|^2 \\ &= \int_{\omega-\delta/2}^{\omega+\delta/2} d\nu D(\nu)|\beta(\nu,\omega,t)|^2 \\ &= D(\omega)\frac{\left(h_0^2 \omega^3 M L^2\right)}{\hbar\pi^4}\int_{\omega-\delta/2}^{\omega+\delta/2} d\nu \frac{\sin^2(\frac{1}{2}t(\nu-\omega))}{(\nu-\omega)^2} \\ &= D(\omega)\frac{\left(h_0^2 \omega^3 M L^2\right)}{\hbar\pi^4}\Xi(t).\end{aligned} \tag{55}$$

Above, $D(\omega)$ is the density of states. The density of states of plane waves per volume $V$ for a given polarization is determined by the mapping,

$$\sum_\nu = \int d\nu D(\nu) = \int d\nu \frac{V\nu^2}{2\pi^2 c^3}. \tag{56}$$

We now consider Fermi's golden rule's limit $t\delta \gg 1$, for which the integral approaches its limiting value

$$\lim_{t\delta \gg 1}\Xi(t) \to \frac{1}{2}\pi t. \tag{57}$$

In this limit

$$P(\omega,t) \approx D(\omega)\frac{\left(h_0^2\omega^3 ML^2\right)t}{2\hbar\pi^3} = \Gamma_{\text{stim}}t, \tag{58}$$

where $\Gamma_{\text{stim}}$ is the rate of stimulated absorption. We have

$$\Gamma_{\text{stim}} = D(\omega)\frac{\left(h_0^2\omega^3 ML^2\right)}{2\hbar\pi^3} = \frac{Vh_0^2\omega^5 ML^2}{4\hbar\pi^5 c^3} = \frac{h_0^2 M v_s^2}{4\hbar\pi^3}. \tag{59}$$

In the above, the factor of volume $V$ from the density of states $D(\omega)$ is taken to be the reduced volume of a single graviton, and we have used a graviton's reduced wavelength to compute this reduced volume as $V = (c/\omega)^3$. Now, we determine the characteristic strain for our detector by requiring that the rate of thermal excitations $\gamma_{\text{th}} = \omega\bar{n}Q^{-1}$ is less than or equal to the rate of stimulated absorption, $\Gamma_{\text{stim}}$ by the monochromatic wavepacket described above. From this, we define the characteristic strain amplitude $h_c$ as the minimum detectable strain amplitude of the wavepacket

$$h_c \equiv 2\pi\sqrt{\frac{\pi k_B T}{M v_s^2 Q}}. \tag{60}$$

Using this standard, we see that minimum detectable strain amplitude $h_0$ for a strictly monochromatic wave is related to the characteristic strain amplitude $h_c$ via the relation $h_c = 2\pi h_0\sqrt{N_c}$ where $N_c$ is the number of cycles observed, which is comparable to how the characteristic strain amplitude $h_c$ for a strictly monochromatic wave is typically defined. We use this result to make the comparison to standard bar detectors, as summarized in the main text and Fig. 2.

**Continuous weak measurement of energy**

Here we describe the continuous measurement of energy of an accoustic mode through Homodyne-like measurements. To derive the measurement operator that describes the continuous measurement of the energy of an acoustic mode, we consider an acoustic mode of frequency $\omega_l$, with the free Hamiltonian[52,71,72]

$$\hat{H}_{\omega_l} = \hbar\omega_l\left(\hat{b}_l^\dagger\hat{b}_l + \frac{1}{2}\right). \tag{61}$$

We consider the probe as a continuous variable system (waveguide photons or a quantum LC circuit) with quadratures $\hat{x},\hat{p}$ (equivalently the charge $\hat{q}$ and phase $\hat{\phi}$ for LC circuit implementations). In the following, we assume that the probe is initialized in a zero-mean Gaussian initial state:

$$|\psi_M\rangle = (2\pi t_m)^{-\frac{1}{4}}\int dx e^{-\frac{x^2}{4t_m}}|x\rangle, \tag{62}$$

with variance $t_m$ in the $x$ basis, that couples to the acoustic mode via the following interaction Hamiltonian:

$$\hat{H}_{\text{int}}^M dt = \sqrt{dt}\hat{p}\hat{N}. \tag{63}$$

The measurement operator is given by

$$\begin{aligned}\hat{M}_{\hat{N}}(y) &= \langle y|e^{-i\hat{H}_{\text{int}}^M dt}|\psi_M\rangle \\ &= (2\pi t_m)^{-\frac{1}{4}}\exp\left[-\frac{(y - \hat{N}\sqrt{dt})^2}{4t_m}\right].\end{aligned} \tag{64}$$

We can re-write the measurement operator in terms of the Homodyne signal $r = y/\sqrt{dt}$ as

$$\hat{M}_{\hat{N}}(r) = (2\pi t_m/dt)^{-\frac{1}{4}}\exp\left\{\left[-\frac{dt(r - \hat{N})^2}{4t_m}\right]\right\}. \tag{65}$$

To describe continuous measurements, we assume that the probe is reset to the initial state at each $dt$. Physically this could mean that, for example, when the probe is a photon, it is a different photon that interacts with the resonator at each instant in time. This represents a continuous stream of single photons that interact with the acoustic mode, and subsequently homodyne detected. For Homodyne sensing of energy of the oscillator, we can represent the readout signal $r(t) \approx \langle\hat{N}(t)\rangle + \sqrt{t_m}\zeta(t)$, where $\zeta(t)$ is a Gaussian white-noise, $\delta$ correlated in time, $\langle\zeta(t)\zeta(t')\rangle = \delta(t - t')$ of variance $1/dt$. Therefore at each instant in time $t$, the density matrix for the acoustic mode can be updated as

$$\rho(t + dt) = \frac{D[dt\beta'(t)]\hat{M}_{\hat{N}}[r(t)]\rho(t)\hat{M}_{\hat{N}}^\dagger[r(t)]D[-dt\beta'(t)]}{\text{tr}\{\hat{M}_{\hat{N}}[r(t)]\rho(t)\hat{M}_{\hat{N}}^\dagger[r(t)]\}}, \tag{66}$$

which describes the quantum evolution of the acoustic mode subject to time-continuous quantum measurements of its energy.

For simulations shown in the main text, we consider the characteristic measurement time $t_m = 2\,\text{s}$. We measure for a duration $t_{\text{meas}} = 40\,\text{s}$, after which we re-initialize the detector to its ground state. We also require that $t_{\text{meas}} < \tau_c$, the coherence time of the acoustic mode for the above-described formalism to apply. In simulations, we confirm that for the optimized mass, maximum absorption occurs in the neighborhood of the resonance for a chirping gravitational wave. Equation (66) can be thought of as the Trotterized representation of the dynamics of an acoustic bar resonator, whose energy is continuously monitored. In the limit of large measurements, $t_{\text{meas}} \gg t_m$, the probability of measuring a single quantum in the acoustic bar resonator approaches its optimum value, $P(1) = 1/e$.

## Data availability
All raw data generated during the current study are available from the corresponding author upon request.

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

## Acknowledgements

We thank Shimon Kolkowitz, Avi Loeb, Stefan Pabst, Michael Tobar, Frank Wilczek, and Ting Yu for the discussions. G.T. thanks the 2022 Research Internship Program at Okinawa Institute of Science and Technology (OIST) for hospitality during the development of this work. This research has made use of data or software obtained from the Gravitational Wave Open Science Center (gwosc.org), a service of the LIGO Scientific Collaboration, the Virgo Collaboration, and KAGRA. This material is based upon work supported by the National Science Foundation under Grant No. 2239498, the European Research Council under Grant No. 742104, the Swedish Research Council under Grant No. 2019-05615, the U.S. Department of Energy, Office of Science, ASCR under Award Number DE-SC0023291, the Branco Weiss Fellowship—Society in Science, the General Sir John Monash Foundation, and the Wallenberg Initiative on Networks and Quantum Information (WINQ). Nordita is partially supported by Nordforsk.

## Author contributions

All authors (G.T., S.K.M., T.B., I.P.) contributed to all aspects of the research with leading input from G.T. and S.K.M. I.P. conceived and supervised the research. All authors (G.T., S.K.M., T.B., I.P.) contributed to writing the manuscript.

## Funding

## Competing interests

The authors declare no competing interests.
