## [Peer Review File · Nature Communications]

Detecting single gravitons with quantum sensingREVIEWER COMMENTS

Reviewer #1 (Remarks to the Author):

In this manuscript the authors consider the possibility of detecting quantum jumps of a cylindrical mechanical resonator from the ground state to the first excited state what are produced by the strain field of passing gravitational wave. Directly detecting a low-energy experimental signature of the quantization of gravity is an interesting idea that has recently gained more attention of the community, including proposals to witness entanglement between macroscopic superpositions that are mediated by the gravitational field (e.g. Refs 5,6,7,8). In this paper the authors point out that the spontaneous emission rate via gravity for the decay of the excited state of a massive resonator is far too slow to be experimentally observed, by considering the stimulated absorption of a passing gravitational field with strain of order 10^{-22} , the expected rate becomes experimentally accessible, of order 1 Hz for a sufficiently massive object. The method relies of the coincidence of detection with that of a large gravitational wave disturbance e.g. by the LIGO detectors. Assuming other excitations that drive the resonator out of the ground state can be suppressed sufficiently, this coincidence can be used to conclude that the gravitational field is responsible for exciting the resonator. A major challenge has to do with the rate of excitation from the thermal background which scales as $kT/(\hbar Q)$.

The idea is novel and interesting, the paper is well written, and I recommend publication. However, the requirements for keeping the thermal excitation rate sufficiently low are quite stringent and appear to be a couple of orders of magnitude beyond the state of the art that has been experimentally achieved so far. The authors might consider for example elaborating more on the existing state of the art. The proposed parameters for example, $T=1\text{mK}$ and $Q=10^{10}$ may be achievable as a long-term goal, however in practice Q factors closer to 10^8 (e.g. Niobe) and temperatures of order 100 mK for large massive bars are more typical. I think this point should be made clearer in the text. Nevertheless, these parameters may be possible in the future and it is useful to point to these requirements as targets for the community.

Reviewer #2 (Remarks to the Author):

This manuscript presents a novel idea to detect single gravitons through the stimulated emission caused by an incident (coherent state) gravitational wave arising from, for example, a gravitational wave.

The basic idea is to consider a massive body that is cooled to the ground state. The authors argue that while the rate of spontaneous emission induced by an incident gravitational wave is extremely small the average rate of stimulated emission can be large and observable. Additionally, the authors argue that, by continuous quantum sensing, they can measure individual quantum jumps in their detector corresponding to individual graviton exchanges.

This work presents an important and novel contribution to the range of near-term quantum gravitational detectors. Other proposed probes of quantum gravity rely on the creation of a coherent, quantum spatial superposition which is experimentally challenging. However, it's important to mention that there have been arguments that such experiments can yield indirect evidence for the existence of quantized radiation (see, e.g., Phys. Rev. D 98, 126009 (2018) and the more rigorous analysis given in Phys. Rev. D 105, 086001 (2022))

Since ground-state cooling can be feasibly achieved with current/near-future technology, the experimental challenge of the approach taken in this paper is the "continuous quantum sensing". It would be nice if the authors expanded a bit more on the technological difficulties/hurdles that one must overcome to achieve this.

In total, I recommend this manuscript for publication with minor revisions.

Reviewer #3 (Remarks to the Author):

The manuscript presents a new idea for detecting single gravitons from GW based on resonant bars. The proposal is extremely interesting and ambitious. But if successful it will be a significant step in solving the most important question of modern physics which is quantum vs classical nature of gravity. The proposal is very timely, it will definitely draw

more attention to the graviton detection problem that has been considered unsolvable. With the recent progress in quantum technologies, optomechanics and related fields, it is definitely the time to reconsider and rethink these ideas.

On the other hand, the result is quite surprising and counterintuitive. Bar GW detectors have been abandoned over 20 years ago due to their inferior sensitivity comparing with laser interferometers. So, can authors describe improvements in terms of strain sensitivity in $1/\sqrt{\text{Hz}}$ required for an old bar detector required for this detection? This is not the same as information in Table 1 which assumes sources. Say we have a new generation of bar detectors, how good should they be comparing to old ones and LIGO?

It would be quite interesting to look into proposal in a source independent way using strain sensitivity. Also, to make this more appealing to experimentalist, it would be nice to present how sensitivity or at least detection probability scales with various parameters, particularly frequency. How fast it degrades if mass is not optimal? Can we even reach optical mass at given frequency? Can multimode operation or two antenna operation be helpful?

Do results depend on boundary conditions? For most bar detectors they do. How sensitivity degrades with going into higher order modes?

The discussion of proof vs evidence is quite interesting. Would we be able to draw any conclusion in case of non-observation of any proposed detections?

Can the scheme be improved by using some interesting quantum states, like ones used in some axion dark matter detectors?

Are authors concern about particle background events, e.g. natural radioactivity and cosmic rays? There are several studies suggesting and observing sensitivity of bar detectors to cosmic rays. It was suggested to move future bar detectors should operate underground like dark matter detectors. The authors suggest to correlate graviton events with LIGO, but this might not be enough. If rate of cosmic ray events is high, there will be certainly one corresponding to each LIGO event. It is certainly easier to detect a muon or alpha particle

than a graviton. So, how sensitive is the proposed systems to these kind of background?

>>> It is evident from equation (20) that this is an integrated effect that builds up across the length of the solid-bar resonator, and therefore represents a unique advantage for detecting gravitons using bar resonators

Not quite evident. Not quite sure what is meant by 'unique advantage'. If this is unique advantage, why it is not discussed in the main text? Or at least I could not find it.

>>> we get the requirement on the detector mass to obtain the maximal transition probability in the presence of $h(t)$:

Does this imply that the system needs tuning to achieve optical performance at each ω ? How important is it to achieve this maximal transition probability?

All methods and methodology used in the manuscript are solid. Literature review is complete, the work contains new original ideas which will bring a lot of attention. It contains all necessary details to make the work reproducible.

Overall, the proposal is extremely interesting, so the manuscript deserves publication in Nature Communications after minor improvements to make it more appealing to experimentalists.

Reviewer #4 (Remarks to the Author):

The idea discussed by the authors is certainly noteworthy and could be suitable for publication to Nature Communications.

I have hesitated on what to suggest because I am a theorist. I understand what the authors do and I do not have any hesitations on the idea discussed and the computations.

However, the noteworthiness of the manuscript mainly lies on its claims that this experiment is feasible, in the 'near' future and that implies it should be just on that basis.

That is, the idea of detecting the analogue of the photoelectric effect for gravity as the authors state correctly is almost as old as the photoelectric effect itself. The likelihood that the effect in principle exists is very high and so searching for it is certainly worthwhile, if this is possible.

There is indeed a significant theoretical point made in this work, the idea to use stimulated emission instead of spontaneous emission, which is certainly a noteworthy idea as the large mass envisioned gives a very significant amplification wrt to Dyson's famous analysis that gravitons can never be detected.

However, I am not sure this would be sufficient unless also the number estimations fall indeed somewhere in the realm of feasible and I could simply not judge myself the feasibility claims made in this paper, I simply do not know the trade on the experimental side.

Personally, if an experimentalist reviewer would say that this experiment or something vaguely along this lines plausible has maybe 10% chance of being realised the next 20 years given current trends, then I would say this is certainly a work worthy of being published in nature communications. I just cannot judge myself whether this is the case or if we are talking about 0.01% :)

Having said this, the manuscript could have done a better job at explaining to non experts in acoustic bar resonators and gravitational wave detectors whether the parameter regime identified in detail stands any chance of being reached and how far we are at the moment, and discuss a bit more what the challenges would be with respect to noise sources. I understand there are length limitations to be considered but I feel the first 2.5 pages of text could go to the point a bit more compactly to make room to discuss the challenges better. It is an extraordinary claim that this could be feasible in the near future, and so this is the claim that more text should be spent to substantiate.

I would thus suggest with the above reservations a revision where the authors are a bit more self critical on the chances of an actual realisation of this experiment.

Manuscript NCOMMS-23-48291-T: Detecting single gravitons with quantum sensing,

by G. Tobar, S. K. Manikandan, T. Beitel and I. Pikovski

Reply to Reviewers

We thank all reviewers for their thorough reading of our manuscript and the many useful and constructive suggestions. In response to the points raised by the reviewers we have expanded our manuscript and hope that the questions and suggestions have been adequately addressed. Below are our point-by-point responses to each reviewer.

We attach a version of the manuscript which has additions and changes highlighted in red, for easier readability. We apologize for the unfortunate delay with the revision, but we hope that the revised manuscript is now suitable for publication.

Reviewer #1 (Remarks to the Author):

The idea is novel and interesting, the paper is well written, and I recommend publication. However, the requirements for keeping the thermal excitation rate sufficiently low are quite stringent and appear to be a couple of orders of magnitude beyond the state of the art that has been experimentally achieved so far. The authors might consider for example elaborating more on the existing state of the art. The proposed parameters for example, $T=1\text{mK}$ and $Q=10^{10}$ may be achievable as a long-term goal, however in practice Q factors closer to 10^8 (e.g. Niobe) and temperatures of order 100 mK for large massive bars are more typical. I think this point should be made clearer in the text. Nevertheless, these parameters may be possible in the future and it is useful to point to these requirements as targets for the community.

We thank the reviewer for the comment. In the revised manuscript we have significantly expanded on this point. Indeed, the required parameters are more challenging than what has been experimentally achieved so far, and we make now a direct comparison to achieved and proposed classical gravitational bar detectors, in particular in the new Figure 2 (with corresponding theory in Appendix B3) and the expanded 'noise' section. We have added concrete results and references to the state-of-the-art for damping and cooling of bar systems. This is to highlight that technological improvements are necessary to achieve our proposed scheme. But as the reviewer points out, we think such improvements are possible in the future and can serve as targets for the development of such systems.

Reviewer #2 (Remarks to the Author):

This work presents an important and novel contribution to the range of near-term quantum gravitational detectors. Other proposed probes of quantum gravity rely on the creation of a coherent, quantum spatial superposition which is experimentally challenging. However, it's important to mention that there have been arguments that such experiments can yield indirect evidence for the existence of quantized radiation (see, e.g., Phys. Rev. D 98, 126009 (2018) and the more rigorous analysis given in Phys. Rev. D 105, 086001 (2022))

We fully agree with the reviewer about the importance of other proposals and the comparison. In the revision we have now included a short discussion on this, including the possible indirect

evidence of virtual gravitons from other works and the suggested reference. This is in the revised 'discussion' section of our manuscript, where we write:

"The graviton detection scheme discussed here, and its experimental requirements, are of a different nature than other proposals for table-top tests of quantum gravity. Such proposals have so far either focused on testing a modified quantum dynamics due to speculative quantum gravity phenomenology [2–5, 10, 11], or entanglement generated by gravitating source masses in superposition [6, 7]. The latter rely on the very challenging creation of macroscopic quantum superpositions. In contrast, in our work the state preparation only relies on ground state cooling, but the difficulty lies in implementing the quantum sensing scheme for massive systems. Our work also focuses on a very different regime of the interaction between matter and gravity, in contrast to other schemes: here we show that exchange of single quanta of energy with gravitational waves can be observed, while proposals to test entanglement generation through gravity focus on the expected Newtonian interaction between static masses in superposition. Arguments can be made that the latter also provide a signature of graviton exchange, but for virtual particles [64] and under additional assumptions [65]. In contrast our work here relies on the direct on-shell exchange of gravitons. Such tests therefore test different but complementary aspects of gravity in the quantum regime."

Since ground-state cooling can be feasibly achieved with current/near-future technology, the experimental challenge of the approach taken in this paper is the "continuous quantum sensing". It would be nice if the authors expanded a bit more on the technological difficulties/hurdles that one must overcome to achieve this.

Indeed, we believe this is the main technical challenge of our proposal, and we have significantly expanded our discussion around the need for technological improvements in the section 'noise' in the main text. We include a comparison to previously achieved and proposed bar detector technology, highlighting what improvements are needed in this regard. We have also included a discussion of continuous quantum sensing, how it could in principle be achieved and a brief comparison to current state-of-the-art, expanding the 'measurement' section. The required non-linear interaction has recently been achieved in an acoustic system (Nat. Phys. 18, 794–799 (2022) – reference 57 in the revised manuscript), albeit in a 16 microgram system. Thus development of sensing capability is needed in order to bridge these several orders of magnitude in mass. However, there are several possibilities that we leave as future research directions - for example, one possibility is to couple light and the fundamental acoustic mode to generate a non-linear coupling. A second possibility is that we couple other propagating acoustic modes such as surface acoustic waves to the fundamental mode and readout surface acoustic waves. A third possibility is to use transducer technology as is common in classical bar detectors to reduce the mass requirements for sensing. Overall, we believe there is a range of possibilities to be explored in future research to address this challenge, and we hope our work motivates such developments.

Reviewer #3 (Remarks to the Author):

On the other hand, the result is quite surprising and counterintuitive. Bar GW detectors have been abandoned over 20 years ago due to their inferior sensitivity comparing with laser interferometers. So, can authors describe improvements in terms of strain sensitivity in $1/\sqrt{\text{Hz}}$ required for an old bar detector required for this detection? This is not the same as information in Table 1 which assumes sources. Say we have a new generation of bar detectors, how good should they be comparing to old ones and LIGO?

We thank the referee for this suggestion, echoing also other reviewers' comments on the need for a more detailed comparison. In the revised manuscript we have added an extensive comparison to classical bar detectors and proposed upgrades, highlighting what improvements are needed to get in the regime that we propose. In essence about 2-3 magnitude of improvement over previous bar detectors would be needed in terms of sensitivity. To make the comparison, we converted the signal to a characteristic strain, as discussed in an added Appendix B3. This essentially measures how well our proposed system would detect classical waves, if it was used to do so. However, for our purposes the optimization we require is different, namely only a single energy transition in the detector. Such operation is not well suited to confirm gravitational waves (which can be done with better means such as with much higher masses). Nevertheless, the comparison is instructive as it shows what noise levels are required and how it compares to previously achieved and proposed bar detectors. That is summarized in a new figure 2.

It would be quite interesting to look into proposal in a source independent way using strain sensitivity. Also, to make this more appealing to experimentalist, it would be nice to present how sensitivity or at least detection probability scales with various parameters, particularly frequency. How fast it degrades if mass is not optimal? Can we even reach optimal mass at given frequency? Can multimode operation or two antenna operation be helpful?

We chose to compare our detector to other detectors by using a characteristic strain h_c for which the detector would be sensitive, as a means of comparison. We performed a careful analysis of this in appendix B3, which also links our theory to previous results in the bar community, with the results fully matching. The sensitivity and its scaling with parameters is now captured in equation B32. For the task of measuring only a single transition though, this is not the best benchmark and instead the probability of excitation is a better signature. If mass is not optimal, it is not that detection is not possible, it's just that the probability for absorption to the $n = 1$ Fock state is not maximized, in this case it may be more likely to transition to an energy state which is not the $n = 1$ Fock state but a higher energy level instead. Therefore, a non-optimal mass doesn't necessarily degrade the detector's sensitivity, but rather it degrades the possibility to measure the $n = 1$ Fock state (transitions to higher energy levels would be fine if a sensor can discern any such level, but this is usually difficult). We added a discussion on its scaling in the appendix B1 below equation B11. The reviewer is also correct in pointing out that multimode operation or several antennas can be helpful, such as for coincidence measurements (also outside the LIGO band) and for better readout. This is indeed a very important and interesting direction to explore, which will likely show how to alleviate some of the challenges outlined in our present manuscript. We leave this question for future dedicated studies on how to better implement our proposed scheme with more concrete quantum bar detector designs.

Do results depend on boundary conditions? For most bar detectors they do. How sensitivity degrades with going into higher order modes?

Indeed our results were derived effectively for a cylindrical bar. The boundary conditions are given in equation A3, and indeed results depend on them, here we use Von Neumann boundary conditions. This enables us to expand the solution as in A4. If we were to use Dirichlet boundary conditions (such that the mode profile decays to zero at the boundaries of the bar), this would lead to a different solution. Thus only particular modes are part of our analysis. For other geometries, the results would also change somewhat and thus also the optimal mass would then be different. For example, more recent bar detectors such as miniGRAIL had focused on spherical designs which has some advantages. In our revision we cite the various designs considered previously. However, our results for the optimal mass would still be within the same order of

magnitude. We leave the optimization of the geometry for a future study. With regards to higher modes, at least for the cylindrical case we consider, the sensitivity indeed degrades with higher modes, scaling as $1/l^2$ in terms of the mode number l . This is highlighted in the interaction Hamiltonian we derived in A16, which includes also higher order mode coupling to GWs.

The discussion of proof vs evidence is quite interesting. Would we be able to draw any conclusion in case of non-observation of any proposed detections?

We thank the reviewer for the query, as we also find the implications of the experiment for quantum gravity particularly interesting and to be explored further. Non-observation is a possible outcome but would be inconsistent with our present expectation of the linearized and quantized interaction between gravitational waves and matter. Without empirical evidence such as in our proposed experiment though, the expected outcomes are just theoretical. The possibility of being surprised in such an experiment is thus very exciting, and highlights the novelty of the regime that would be probed in such an experiment. We added a small discussion on this at the end of the discussion section, emphasizing that outcomes other than what we compute are possible and might indicate new physics: *"We note that the observation of the exchange of single quanta between gravitational waves and matter may also offer surprises and proceed differently from the expected behavior described here, and empirical experimental evidence is highly desirable even if the linearized limit of the interaction is well understood theoretically"*

Can the scheme be improved by using some interesting quantum states, like ones used in some axion dark matter detectors?

This is a good question and we are not sure at this stage. Likely there are quantum technology improvements one can consider and it is worth exploring further in the future. We have included a reference to a study on using Rydberg atoms for graviton absorption [26] – while not competitive with the rates that collective modes provide, it shows that quantum states could offer interesting modifications (in this case to Weinberg's original computation of atomic processes in the presence of gravitational waves).

Are authors concern about particle background events, e.g. natural radioactivity and cosmic rays? There are several studies suggesting and observing sensitivity of bar detectors to cosmic rays. It was suggested to move future bar detectors should operate underground like dark matter detectors. The authors suggest to correlate graviton events with LIGO, but this might not be enough. If rate of cosmic ray events is high, there will be certainly one corresponding to each LIGO event. It is certainly easier to detect a muon or alpha particle than a graviton. So, how sensitive is the proposed systems to these kind of background?

The referee is exactly right that noise sources such as cosmic rays will be significant challenges. We have added a discussion on this and other noise sources in the section on 'noise', which also are relevant for classical bar detectors. Cosmic ray sensors will indeed be required, as cosmic showers would excite the bar and such events need to be excluded. Mitigating such noise sources will require similar strategies as developed for bar detectors, but as our new Figure 2 shows, operating our detector essentially means mitigating the noise better by about 2 orders of magnitude. However, we note that most bars had the development of quantum limited sensing in mind, which would achieve the precision we require here. Thus we believe that the mitigation of noise is possible, albeit challenging. The short window of detection also means that correlation measurements should provide significant improvement over non-continuous effects, but the success probability would overall be degraded.

>>> *It is evident from equation (20) that this is an integrated effect that builds up across the length of the solid-bar resonator, and therefore represents a unique advantage for detecting gravitons using bar resonators*

Not quite evident. Not quite sure what is meant by 'unique advantage'. If this is unique advantage, why it is not discussed in the main text? Or at least I could not find it.

The referee is correct and our wording was both sloppy and ambiguous. We wanted to highlight that going from atoms to collective systems provides significant enhancement, while still being able to operate in the quantum regime. In our appendix A we show how this is realized in mathematical detail, with the various contributions captured in eqs. A14 and A15. We have changed the wording in Appendix A to reflect this more clearly, now reading below eq. A14: "*Here the sum over all atoms results in an enhanced effect that builds up across the length of the resonator, scaling with mass and length*".

>>> *we get the requirement on the detector mass to obtain the maximal transition probability in the presence of $h(t)$:*

Does this imply that the system needs tuning to achieve optical performance at each omega? How important is it to achieve this maximal transition probability?

Yes the reviewer is right – the mass of the bar is optimized only for a particular GW frequency and profile. It would indeed be very desirable for the bar to be tunable, but this is rare and to our knowledge only realistically possible with superfluid Helium detectors where the resonance frequency (and speed of sound) can be tuned through pressurization. In the main section on 'compact binary mergers', we have added the sentence "*In terms of material, a higher speed of sound is desirable, but other features can make other materials more favorable, such as the tunability of the resonance frequency in superfluid Helium detectors*" to highlight this point. However, in most cases a specific realization would be ideally sensitive only to one type of GW source, thus we envision an optimized design that would be sensitive to nearby neutron-star mergers, with masses around 20kg. We also note that we added beryllium in our Table I as a promising candidate, which has also previously been considered as a potential upgrade to the miniGRAIL detector.

With regards to maximizing the probability, this is conceptually not essential: we simply want to show how to achieve the best physical parameters. Single transitions would also occur for unoptimized parameters, but with reduced probability. The main goal is to be able to resolve a discrete transition of energy, and this could in principle be achieved also with somewhat different physical parameters – the scaling is given in equation B10 with added discussion below it. Thus the optimization is somewhat flexible, but it shows the needed orders of magnitude for the physical parameters. We have added a statement for clarification in the main text: "*A higher mass would increase the strain sensitivity, but would result in excitations distributed among predominantly higher energy levels in the resonator.*", and "*Given otherwise fixed system parameters, the rate and excitation probability increase with the resonant frequency.*"

All methods and methodology used in the manuscript are solid. Literature review is complete, the work contains new original ideas which will bring a lot of attention. It contains all necessary details to make the work reproducible.

Overall, the proposal is extremely interesting, so the manuscript deserves publication in Nature

Communications after minor improvements to make it more appealing to experimentalists.

We thank the referee for the positive assessment. We hope the changes do highlight more clearly both the challenges ahead and the comparison to the state-of-the-art. Ultimately, dedicated research will be needed to make such an experiment a reality, and we hope to do so in collaboration with experimental groups in the future.

Reviewer #4 (Remarks to the Author):

However, the noteworthiness of the manuscript mainly lies on its claims that this experiment is feasible, in the `near' future and that implies it should be just on that basis.

That is, the idea of detecting the analogue of the photoelectric effect for gravity as the authors state correctly is almost as old as the photoelectric effect itself. The likelihood that the effect in principle exists is very high and so searching for it is certainly worthwhile, if this is possible.

There is indeed a significant theoretical point made in this work, the idea to use stimulated emission instead of spontaneous emission, which is certainly a noteworthy idea as the large mass envisioned gives a very significant amplification wrt to Dyson's famous analysis that gravitons can never be detected.

We fully agree with the referee that the true noteworthiness is with the result that such single graviton detection events could be observed in realistic experiments. We allow us to summarize our reasoning here on what we believe our work adds to our current knowledge of the effect. As the reviewer points out, the photoelectric effect for gravity has been considered numerous times and the standard results are well known, including in the classic book by Weinberg. We believe our main contribution is to show that the problem can be addressed from a new perspective, which relies on several modifications. In essence, these are (i) considering quantum systems that are macroscopic and not on atomic scales, (ii) considering stimulated emission as we now have LIGO detections to cross-correlate and confirm background gravitational waves that can cause such a stimulated process, and (iii) consider quantum sensing to confirm discrete transitions in energy that mimic atomic spectra. These do not alter the fundamental understanding of the gravitational photoelectric effect which follows directly from the quantized gravity-matter coupling as we know it from linearized gravity, but rather show that new ideas (inspired by new experimental capabilities) can circumvent the challenges that were seen as impossible to overcome.

However, I am not sure this would be sufficient unless also the number estimations fall indeed somewhere in the realm of feasible and I could simply not judge myself the feasibility claims made in this paper, I simply do not know the trade on the experimental side.

Personally, if an experimentalist reviewer would say that this experiment or something vaguely along this lines plausible has maybe 10% chance of being realised the next 20 years given current trends, then I would say this is certainly a work worthy of being published in nature communications. I just cannot judge myself whether this is the case or if we are talking about 0.01% :)

We understand the reasoning of the reviewer and share in spirit this perception. Our excitement about our result is that we believe that our scheme is a first realistically achievable route for the confirmation of the gravitational photo-electric effect. There is much uncertainty as with any large-scale effort for entirely new experiments, but we are very optimistic about the prospects especially

in comparison to the previous perception that it's simply impossible. Coming from the community that explores low-energy searches for quantum gravity phenomenology, we believe our proposal is not more difficult, and maybe even simpler, than some of the proposed alternative schemes to detect quantum signatures of gravity (they typically require macroscopic superpositions that are exceptionally hard to maintain, none have considered the gravitational photoelectric effect so far). We have added in the revised manuscript in the 'discussion' section a short comparison to other low-energy proposals that are of different nature, and we have now also significantly expanded our discussion of the experimental improvements needed to achieve our proposal, such as the added figure 2 and the expanded sections on 'noise' and 'measurement'. Nevertheless, our proposal is of course still of theoretical nature, but we strongly believe that it offers the first realistic path for an experiment of this kind.

Having said this, the manuscript could have done a better job at explaining to non experts in acoustic bar resonators and gravitational wave detectors whether the parameter regime identified in detail stands any chance of being reached and how far we are at the moment, and discuss a bit more what the challenges would be with respect to noise sources. I understand there are length limitations to be considered but I feel the first 2.5 pages of text could go to the point a bit more compactly to make room to discuss the challenges better. It is an extraordinary claim that this could be feasible in the near future, and so this is the claim that more text should be spent to substantiate.

I would thus suggest with the above reservations a revision where the authors are a bit more self critical on the chances of an actual realisation of this experiment.

We thank the reviewer for the suggestion, which also mirrors similar suggestions by the other reviewers. We have added now a much more comprehensive comparison to the state-of-the-art in different communities, with a large expansion on the comparison to previously achieved and proposed classical bar detectors (the new figure 2 summarizes that, with the theory in Appendix B3). We believe these additions significantly improve the manuscript and also further clarify that we do not wish to understate the big challenges for an actual experiment. Our excitement for the proposal stems from the previous widespread perception in the physics community that such type of experiments might never be achievable at all, and so our optimistic outlook is inspired by having found a new promising path with essentially three new ideas as described above. Nevertheless, a real experiment would require improvements of bar detector sensitivity by about 2-3 orders of magnitude and also the scaling up of QND measurement capabilities to larger masses than what has been achieved so far (most recently in 2022 for 16 micrograms, our reference 57 in the revised manuscript). Nevertheless, there has been tremendous and steady experimental improvement especially on the quantum state preparation and sensing part, and thus we remain very optimistic that dedicated efforts could achieve such gravitational photoelectric experiments. But more technology development (and good new ideas) will be required. We hope the revisions in our manuscript reflect this honest assessment, while highlighting the novelty of our proposed approach.

REVIEWERS' COMMENTS

Reviewer #1 (Remarks to the Author):

The authors have made significant revisions in response to my comments and the comments of the other referees which have improved the manuscript, including a discussion detailing the sensitivity requirements and comparing with the state of the art in bar detector performance. In particular, the new Figure 2 and appendix B.3 as well as the amended Table 1 and the new text on pages 11-13 do a good job clarifying the related questions in my first report as well as the questions raised by other reviewers, and I am in favor of publication of the paper in its present form.

Reviewer #2 (Remarks to the Author):

The authors have adequately addressed my comments. The generation of entanglement via Newtonian interactions can only yield indirect arguments regarding the existence/properties of the gravitons. The proposed experiment goes further than this since it directly involves on-shell gravitons. Additionally, the expanded discussion in the "noise" section and "measurement" sections of the paper adequately addresses my concerns regarding the experimental feasibility/challenges of the experiment.

I recommend this paper for publication.

Reviewer #3 (Remarks to the Author):

The authors have responded to all my previous questions and concerns. I recommend this article for publication in Nature Communications.